# OPC: ONE-POINT-CONTRACTION UNLEARNING TOWARD DEEP FEATURE FORGETTING

## ABSTRACT

Machine unlearning seeks to remove the influence of specific data or classes from trained models to meet privacy or legal requirements. However, existing methods often achieve only shallow forgetting: while outputs change, internal representations still retain enough information to reconstruct the forgotten data or behavior. We demonstrate this vulnerability via feature and data reconstruction attacks, showing that most unlearned features remain informative enough to recover both model performance and raw inputs from the forget set. To address this issue, we propose **OPC** (One-Point Contraction), a simple yet effective unlearning method that contracts the output representations of forget data toward the origin. By limiting representational capacity to a single point, OPC selectively erases feature-level information associated with the forget set. Empirical evaluations on image classification benchmarks show that OPC achieves strong unlearning efficacy and superior robustness against recovery and reconstruction attacks. We further extend OPC to generative diffusion models, validating its effectiveness in the context of conditional image generation. Applied to Stable Diffusion, OPC enables fine-grained removal of concept-level information, achieving state-of-the-art performance in generative unlearning. These results demonstrate OPC's broad applicability and its potential for precise, task-aware control of forgetting across both discriminative and generative domains.

## 1 INTRODUCTION

Machine unlearning, with the aim of selectively removing the influence of specific data instances on a given model without requiring full retraining of the model (Cao & Yang, 2015), has emerged as a significant research frontier in deep learning (Shaik et al., 2024). The quest for effective and efficiency methods to make models "forget" addresses technical demands for excising outdated or erroneous data and legal compliance to recent privacy mandates such as the General Data Protection Regulation (GDPR). However, existing methods of machine unlearning such as (Fan et al., 2024; Thudi et al., 2022; Golatkar et al., 2020; Kurmanji et al., 2023) fail to make models "forget" the internal feature representations of forgotten data. The residual information can be exploited to pose privacy risks, failed compliance, and even adversarial attacks to reverse the unlearning itself.

The threat is real. Membership inference attacks (Shokri et al., 2017) on a given model demonstrated that latent feature representations can leak information on whether individual data is used in training the model. Moreover, recent reconstruction attacks (Bertran et al., 2024; Hu et al., 2024) successfully recover the data "forgotten" by the unlearned models, thereby exposing the risk of shallow unlearning by many existing approaches.

Hence we raise a pivotal question: *can machine unlearning allow models to forget beyond recovery?* Answering yes to this question will contribute to research for theoretically well-founded robust unlearning of deep learning based models. In this work, we make four key contributions to answer this question positively:

- Establish a theoretical foundation of how to achieve "deep feature forgetting".

- Propose a novel unlearning algorithm, named **OPC** unlearning, based on one-point-contraction (OPC) strategy theoretical uncertainty in feature representations.

- Comprehensive empirical validation of the effectiveness of OPC, demonstrating that OPC-unlearned model forgets much deeper than 12 existing machine unlearning methods.
- Verifying generalizability of OPC by applying it to generative diffusion models with state-of-the-art performance on diffusion unlearning benchmark.

# 2 RELATED WORKS

## 2.1 MACHINE UNLEARNING (MU)

MU focuses on efficiently removing the influence of specific data, the *forget set*, from trained models, which is important for data privacy, user consent withdrawal, and regulatory compliance (e.g., GDPR's "right to be forgotten") GDPR. Methods typically aim to erase the forget set while preserving performance on the *retain set*. Representative approaches are summarized below, with details in Section B.

- **Classification Unlearning**: Such as GA (Thudi et al., 2022), RL (Golatkar et al., 2020), BE (Chen et al., 2023), FT (Warnecke et al., 2023), NGD (Chourasia & Shah, 2023), NegGrad+ (Kurmanji et al., 2023), EUk & CFk (Goel et al., 2022), SCRUB (Kurmanji et al., 2023), and BT (Chundawat et al., 2023), $l1$-sparse (Jia et al., 2023).
- **Diffusion Unlearning**: Such as EDiff (Wu et al., 2025), ESD (Gandikota et al., 2023), FMN (Zhang et al., 2024a), SHS (Wu & Harandi, 2024), CA (Kumari et al., 2023), SEOT (Li et al., 2024), SPM (Lyu et al., 2024), SAeUron (Cywiński & Deja, 2025) and UCE (Gandikota et al., 2024).
- **Cross-domain Unlearning**: Applicable methods across both domains, including SalUn (Fan et al., 2024).

## 2.2 ATTACKS ON MU

MU is vulnerable to adversarial attacks. Membership inference attacks (MIA) (Shokri et al., 2017) can reveal whether forget-set data still resembles the training or test set, indicating unlearning success.

Reconstruction-based attacks are even more threatening, as latent features can be exploited to recover forgotten data. Inversion-based methods (Hu et al., 2024) align gradients from GA-unlearned models to reconstruct forget-set samples, highlighting limitations of shallow unlearning.

We applied this inversion attack to benchmark scenarios. As shown in Fig. 1, many methods leaked forget-set information, while OPC effectively resisted recovery. Additional setup and results are in Section D.3.

## 2.3 FEATURE MAGNITUDE AND OOD

In literature of transfer learning and OOD-detection, the role of feature norm was observed empirically and employed in practice, that the features of OOD data are observed to have smaller magnitudes (Dhamija et al., 2018; Tack et al., 2020; Huang et al., 2021) and thus able to be distinguished. This phenomenon is explained theoretically in Park et al. (2023) that the feature norms can be considered as a confidence value of a classifier. Motivated by the role of feature norm, (Yuan et al., 2017; Xu et al., 2019; Kumar et al., 2023) maximize the feature norm for better performance and transferability.

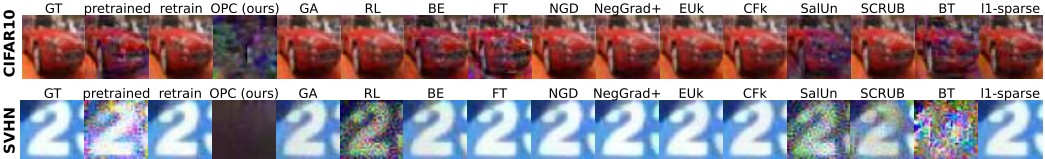

Figure 1: The results of unlearning inversion attack. GT represents the ground truth image from the forget set of each dataset and others are the results from each unlearned model.

# 3 DEEP FEATURE FORGETTING WITH ONE-POINT-CONTRACTION

In this work, we introduce concept of deep and shallow forgetting in Section 3.1 and propose novel MU method OPC in Section 3.2 which aim to seek deep forgetting.

Within this paper, we denote $\mathcal{D}$ be the full dataset, partitioned into four disjoint subsets: $\mathcal{D}_r, \mathcal{D}_f, \mathcal{D}_{val}, \mathcal{D}_{test}$ which are retain set, forget set, validation set and test set respectively.

We assume the model $\mathbf{m}_\theta$ follows the standard encoder–predictor structure $\mathbf{m}_\theta = g_\theta \circ f_\theta$, where $f_\theta$ is the feature extractor and $g_\theta$ the prediction head or diffusion denoiser. This decomposition allows us to analyze changes in learned feature representations independently of the classification layer.

## 3.1 DEEP FEATURE FORGETTING

As listed in Section 2.2 and Fig. 1, there are attacks against MU methods, revealing vulnerabilities to privacy leakage, which indicates that unlearned models still produce informative features on the forget samples.

The conventional metrics of MU, which are mostly logit-level, are not capable of detecting this vulnerability, as MU baselines with strong performance were still vulnerable. Instead, it is worth considering the feature level, whether information about the unlearn target still survives, since practitioners often exploit pretrained model encoders for transfer learning or distillation.

We found that many MU methods exhibit *shallow forgetting*, where the model's predictions on the forget set degrade but the underlying features still encode meaningful information, leaving the model vulnerable to recovery attacks that reconstruct forgotten data from the unlearned model.

In contrast to shallow forgetting, we propose a stricter goal for MU: to completely eliminate the detailed information content of the forget set from the model's internal representations. We define this as *deep forgetting*, where the learned features of the unlearned model are no longer informative about the forgotten data, making the model resistant to data leakage attacks.

## 3.2 OUR METHOD: ONE-POINT-CONTRACTION

We propose One-Point Contraction (**OPC**), a simple yet effective approach for MU that contracts the feature representations of forget samples into single point, the origin 0. This idea stems from two insights: (1) a single point and its local neighborhood have inherently limited representational capacity, and (2) forgotten samples should yield low-norm logits indicative of high uncertainty, in line with how OOD samples behave.

We implement the contraction as an optimization problem to minimize the $\ell_2$ norm of the logits $\mathbf{m}_\theta(x)$ for the forget samples $x \in \mathcal{D}_f$, while preserving performance on retain samples via the standard cross-entropy loss. The unlearning process would be performed by minimizing the following loss function represents the heart of OPC unlearning:

$$\mathcal{L}_{OPC} = \mathbb{E}_{x,y \sim \mathcal{D}_r} \mathcal{L}_{CE}(\mathbf{m}_\theta(x), y) + \mathbb{E}_{x,y \sim \mathcal{D}_f} \|\mathbf{m}_\theta(x)\|_2. \tag{1}$$

The core idea of **OPC**, forcing forget-set feature vectors to have small norms, is closely related to prediction uncertainty. Ideally, unlearned data should be treated as unseen (OOD) samples, leading the model to exhibit high uncertainty with small feature norms. We formalize this relationship in the following theorem, establishing a lower bound on the predictive entropy as a function of feature norm.

**Theorem 3.1.** *Let $C$ be number of classes. Suppose $\mathbf{h} = \mathbf{m}_\theta(x) \in B_r(0)$ where $B_r(0)$ is the ball of radius $r$ centered at origin. Then the entropy $H(softmax(\mathbf{h}))$ of predicted probability has following lower bound parameterized by $r$ and $C$:*

$$H^*(r, C) := \min_{\mathbf{h} \in B_r(0)} H(softmax(\mathbf{h})) > \log \left( 1 + (C-1) \exp \left( -\sqrt{\frac{C}{C-1}} r \right) \right) \tag{2}$$

*Proof of Theorem 3.1.* The exact formula of $H^*(r, C)$ is given by

$$H^*(r, C) = \log\left(1 + \frac{1}{\kappa}\right) + \frac{\log(\kappa(C-1))}{\kappa+1},\tag{3}$$

where $\kappa = \frac{1}{C-1}\exp\left(\sqrt{\frac{C}{C-1}}r\right)$ and $\log\left(1 + \frac{1}{\kappa}\right)$ is equal to RHS of Eq. (2). For the proof of the exact formula, we state that the space of low-entropy features and the ball $B_r(0)$ shows geometric mismatch in $\mathbf{q}$-space, where $\mathbf{q} = \exp(\mathbf{h})$. Therefore, if $r$ is small then no element in $B_r(0)$ can have small entropy and confidently predicted. Detailed proof is in Section A. □

As the feature norm $r$ decreases, the exponential term $\exp\left(-\sqrt{\frac{C}{C-1}}r\right)$ approaches 1, pushing the lower bound in Eq. (2) toward $\log(C)$, the maximum possible entropy. Conversely, as $r$ increases, the lower bound decreases, reflecting that more confident predictions become available.

## 4 EXPERIMENTS ON CLASSIFICATION

We systematically evaluate unlearning methods in terms of feature forgetting and vulnerability using image classification benchmarks. Feature forgetting is quantified via CKA in Section 4.2, measuring the similarity between pretrained and unlearned representations. We further assess whether unlearned features can be recovered through linear transformation attacks in Section 4.3, revealing potential vulnerabilities in the forgetting process.

Overall unlearning performance is presented in Section 4.4, showing that many methods achieve high scores on standard metrics despite exhibiting only shallow forgetting. This underscores a critical limitation of current evaluation metrics, which may overestimate unlearning effectiveness and fail to capture whether sensitive information has truly been removed.

### 4.1 EXPERIMENT SETTINGS

We evaluate unlearning methods on CIFAR10 and SVHN using ResNet-18, considering two scenarios: class unlearning, where $\mathcal{D}_f$ contains classes 0, 1, and 2 (30% of classes), and random unlearning, where 10% of training samples are randomly selected. Additional results are in Section E.

Unlike many existing works that aim to approximate a retrained model, our evaluation policy seeks to maximize forgetting of $\mathcal{D}_f$ while preserving performance on the retain set $\mathcal{D}_r$ and test set $\mathcal{D}_{test}$. We do not prematurely stop unlearning when $\mathcal{D}_f$ performance drops below that of a retrained model, as long as the retained utility remains unaffected.

### 4.2 CKA: FEATURE SIMILARITY MEASUREMENT

We investigate the similarity between pretrained and unlearned features to better understand their representational alignment. For the quantitative analysis, we exploit CKA Cortes et al. (2012); Kornblith et al. (2019) measurement with Kim & Han (2023) implementation, to measure the similarity between unlearned features and pretrained features. Note that the CKA is invariant under scaling and orthogonal transformation, which allows the measurement between distinct models, disregarding the magnitude of the feature.

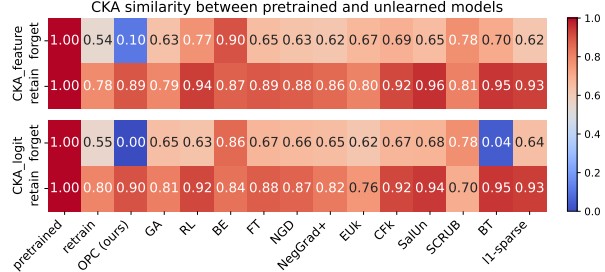

Figure 2: CKA similarity between the pretrained and unlearned models on CIFAR10 (30% class unlearning). CKA-feature and CKA-logit indicate scores computed on $f_\theta(x)$ and $\mathbf{m}_\theta$, respectively.

The results are visualized in Fig. 2. On forget dataset, we could achieve near-zero similarity compared to the original features and logit with OPC, while most of benchmark methods remains to be similar. We may consider this low similarity

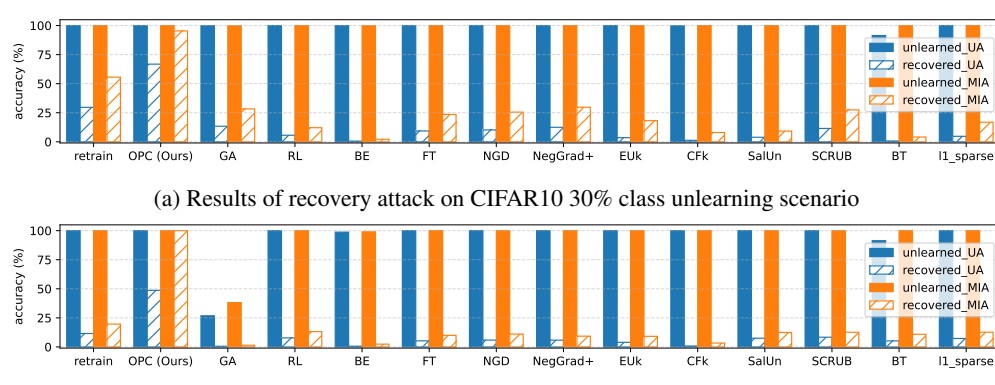

(a) Results of recovery attack on CIFAR10 30% class unlearning scenario

(b) Results of recovery attack on SVHN 30% class unlearning scenario

Figure 3: UA and MIA score of unlearned model and FM-recovered model.

as a direct evidence of deep feature forgetting. For the retain set, the retain features from our method and others show high similarity, which implies that OPC unlearning did not harm the models' ability on the retain dataset.

### 4.3 RECOVERY VIA FEATURE MAPPING

As shown in Section 4.2, pretrained and unlearned forget features are strongly correlated. We further explore whether a linear transformation can map unlearned features back to pretrained ones, which would indicate that unlearning mainly affects the prediction head.

To find the weight matrix $W^*$ that maps the unlearned features to the pretrained features, we formulate the following ordinary least squares problem:

$$W^* = \arg\min_{W} \sum_{x \in \mathcal{D}} \| f_{\theta^0}(x) - W f_{\theta^{un}}(x) \|_2^2, \tag{4}$$

where $\mathcal{D}$ is a sample dataset, and $\theta^0$ and $\theta^{un}$ are the pretrained and unlearned parameters, respectively.

After obtaining $W^*$ by solving linear least square problem, we recover the pretrained feature by multiplying $W^*$ to unlearned feature. We denote this recovery as FM (feature mapping) recovery, where recovered feature can be written as $W^* f_{\theta^{un}}(x)$. We evaluate FM-recovered features using pretrained head $g_{\theta^0}$ and external decoder in subsequent sections. Surprisingly, almost all MU methods were severely vulnerable under this simple attack which doesn't require access to the train data or gradient information.

#### 4.3.1 PERFORMANCE RECOVERY

We use pretrained classifier head $g_{\theta^0}$ to measure the performance of recovered features. The recovered model is represented as $g_{\theta^0} \circ W^* \circ f_{\theta^{un}}$.

Fig. 3 presents the unlearned accuracy (UA), $1 - (\text{accuracy on } \mathcal{D}_f)$ and $\textbf{MIA}^e$ (mia efficacy), under a FM recovery attack. The detailed numbers of recovered performance including accuracies on each dataset, and the $\textbf{MIA}$ scores can be found in Table D.2, in Section D.

Our results reveal that nearly all baseline unlearning methods are vulnerable to this attack: their UA and $MIA^e$ drops substantially, indicating that a considerable portion of the forgotten performance on $\mathcal{D}_f$ can be recovered with minimal effort. Surprisingly, even the retrained model exhibits non-trivial recovery, though it remains more resistant than most unlearning baselines.

In contrast, our proposed method, OPC, demonstrates strong robustness to this recovery attack. On CIFAR10 with class unlearning, the UA remains near 70%, which aligns with the expected UA of random classifier. This robustness arises from OPC's one-point contraction toward the origin, collapsing features to a non-informative point that resists linear reconstruction.

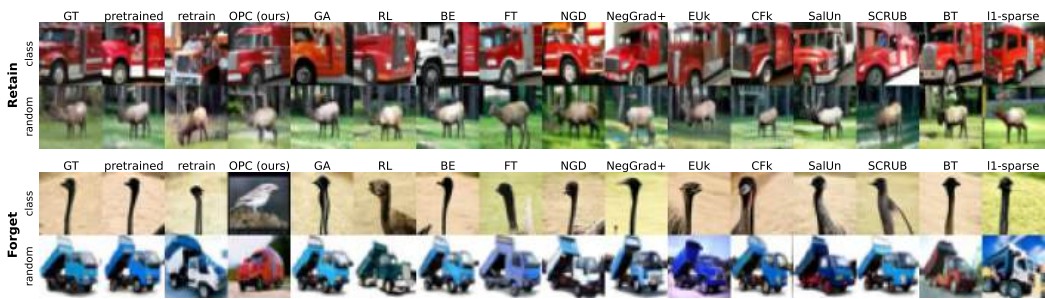

Figure 4: The results of DDPM decoder reconstruction. The target images are sampled from the $\mathcal{D}_r$ and $\mathcal{D}_f$ under both CIFAR10 30% class and 10% random unlearning scenario. GT represents the ground truth image from the dataset and others are the results of reconstruction from each unlearned model.

Table 1: Unlearning performance on 30% Class unlearning scenario

| CIFAR10 | Train $\mathcal{D}_f$ | Train $\mathcal{D}_r$ | Test $\mathcal{D}_f$ | Test $\mathcal{D}_r$ | MIA$^e$ | SVHN | Train $\mathcal{D}_f$ | Train $\mathcal{D}_r$ | Test $\mathcal{D}_f$ | Test $\mathcal{D}_r$ | MIA$^e$ |
|---|---|---|---|---|---|---|---|---|---|---|---|
| Pretrained | 99.444 | 99.416 | 94.800 | 94.400 | 0.015 | Pretrained | 99.531 | 99.172 | 94.960 | 91.110 | 0.009 |
| Retrain | 0.000 | 99.981 | 0.000 | 91.700 | 1.000 | Retrain | 0.000 | 99.997 | 0.000 | 92.440 | 1.000 |
| **OPC** (ours) | 0.000 | 99.606 | 0.000 | 93.143 | 1.000 | **OPC** (ours) | 0.011 | 99.612 | 0.009 | 94.142 | 1.000 |
| GA (Thudi et al., 2022) | 0.148 | 87.771 | 0.033 | 84.057 | 0.998 | GA (Thudi et al., 2022) | 73.220 | 96.477 | 62.618 | 86.270 | 0.381 |
| RL (Golatkar et al., 2020) | 0.000 | 99.060 | 0.000 | 93.529 | 1.000 | RL (Golatkar et al., 2020) | 0.000 | 99.997 | 0.000 | 93.876 | 1.000 |
| BE (Chen et al., 2023) | 0.037 | 93.168 | 0.000 | 85.214 | 0.998 | BE (Chen et al., 2023) | 1.240 | 95.355 | 0.910 | 78.690 | 0.990 |
| FT (Warnecke et al., 2023) | 0.000 | 98.994 | 0.000 | 93.457 | 1.000 | FT (Warnecke et al., 2023) | 0.034 | 99.997 | 0.009 | 94.535 | 1.000 |
| NGD (Chourasia & Shah, 2023) | 0.000 | 98.498 | 0.000 | 93.071 | 1.000 | NGD (Chourasia & Shah, 2023) | 0.000 | 99.997 | 0.000 | 94.854 | 1.000 |
| NegGrad+ (Kurmanji et al., 2023) | 0.000 | 98.638 | 0.000 | 93.014 | 1.000 | NegGrad+ (Kurmanji et al., 2023) | 0.000 | 97.997 | 0.000 | 91.642 | 1.000 |
| EUk (Goel et al., 2022) | 0.000 | 99.616 | 0.000 | 94.629 | 1.000 | EUk (Goel et al., 2022) | 0.000 | 99.997 | 0.000 | 92.826 | 1.000 |
| CFk (Goel et al., 2022) | 0.170 | 99.759 | 0.167 | 94.929 | 1.000 | CFk (Goel et al., 2022) | 0.000 | 99.997 | 0.000 | 92.945 | 1.000 |
| SalUn (Fan et al., 2024) | 0.000 | 99.743 | 0.000 | 94.786 | 1.000 | SalUn (Fan et al., 2024) | 0.000 | 99.990 | 0.000 | 93.910 | 1.000 |
| SCRUB (Kurmanji et al., 2023) | 0.000 | 98.060 | 0.000 | 93.457 | 1.000 | SCRUB (Kurmanji et al., 2023) | 0.008 | 94.995 | 0.000 | 89.129 | 1.000 |
| BT (Chundawat et al., 2023) | 8.578 | 99.502 | 7.533 | 95.286 | 1.000 | BT (Chundawat et al., 2023) | 8.633 | 99.210 | 4.904 | 93.437 | 1.000 |
| $l1$-sparse (Jia et al., 2023) | 0.000 | 99.425 | 0.000 | 94.386 | 1.000 | $l1$-sparse (Jia et al., 2023) | 0.000 | 98.954 | 0.000 | 92.872 | 1.000 |

### 4.3.2 IMAGE RECONSTRUCTION VIA DDPM DECODER

Beyond the class information, we suspect more information is retained on forget feature after unlearning. To check how the unlearned features are informative, we applied FM-recovery and further evaluate the recovered feature qualitatively using generative decoder, which is a generative model trained on pretrained features to recover the input image.

In implementation of generative decoder, we exploit DDPM (Ho et al., 2020) and train it using train dataset, to generate image $x$ conditioned by pretrained feature $f_{\theta^0(x)}$.

The results in Fig. 4 show that, while the generative decoder produces reconstructions slightly different from the original images, important details are preserved. For retain data, all unlearning methods leave features largely unchanged, maintaining input information. In contrast, for forget data, only OPC consistently removes class information and feature-level details, whereas most other methods preserve them. This highlights the shallow forgetting common in MU: even when UA and MIA indicate success, most methods fail to truly erase information at the feature level.

## 4.4 UNLEARNING PERFORMANCE

As observed in previous sections, most existing unlearning methods fail to sufficiently remove learned information at the feature level. Here, we validate that unlearned models with shallow forgetting and vulnerability are still effective under logit-based evaluations. Performance is measured using accuracies on $\mathcal{D}f$, $\mathcal{D}r$, and $\mathcal{D}test$, along with the MIA-efficacy score **MIA**$^e$, which quantifies unlearning success. For the class unlearning scenario, $\mathcal{D}test$ is further split into test $\mathcal{D}_f$ and test $\mathcal{D}_r$, and for element unlearning, we introduce the MIA-privacy score **MIA**$^p$ to assess privacy risk. Higher **MIA**$^e$ and **MIA**$^p$ indicate successful unlearning and greater privacy risk, respectively Jia et al. (2023).

For the class unlearning scenario, the results on both CIFAR10 and SVHN are listed in Table 1. With the exception of GA and BT, most methods succeeded to reduce the accuracy on $\mathcal{D}_f$ while preserving the accuracy on $\mathcal{D}_r$. The **MIA**$^e$ score also shows the unlearning was successfully performed.

The results on random forgetting can be found in Table 2. While most methods failed to reduce the accuracy on $\mathcal{D}_f$ below that of the retrained model, likely due to their stronger generalization ability,

Table 2: Unlearning performance on 10% random unlearning scenario

| CIFAR10 | $\mathcal{D}_f$ | $\mathcal{D}_r$ | $\mathcal{D}_{test}$ | $\mathbf{MIA}^e$ | $\mathbf{MIA}^p$ | SVHN | $\mathcal{D}_f$ | $\mathcal{D}_r$ | $\mathcal{D}_{test}$ | $\mathbf{MIA}^e$ | $\mathbf{MIA}^p$ |
|---|---|---|---|---|---|---|---|---|---|---|---|
| Pretrained | 99.356 | 99.432 | 94.520 | 0.015 | 0.545 | Pretrained | 99.151 | 99.334 | 92.736 | 0.015 | 0.563 |
| Retrain | 90.756 | 99.995 | 90.480 | 0.149 | 0.577 | Retrain | 92.947 | 99.998 | 92.490 | 0.154 | 0.583 |
| **OPC** (ours) | 84.244 | 99.190 | 90.930 | 0.627 | 0.570 | **OPC** (ours) | 7.493 | 99.949 | 92.636 | 1.000 | 0.607 |
| GA(Thudi et al., 2022) | 99.267 | 99.435 | 94.340 | 0.018 | 0.544 | GA(Thudi et al., 2022) | 98.832 | 99.280 | 92.190 | 0.016 | 0.564 |
| RL(Golatkar et al., 2020) | 93.356 | 99.948 | 93.680 | 0.272 | 0.570 | RL(Golatkar et al., 2020) | 92.492 | 97.075 | 92.002 | 0.227 | 0.534 |
| BE(Chen et al., 2023) | 99.378 | 99.440 | 94.480 | 0.016 | 0.545 | BE(Chen et al., 2023) | 99.029 | 99.134 | 90.854 | 0.029 | 0.580 |
| FT(Warnecke et al., 2023) | 95.267 | 99.694 | 92.890 | 0.082 | 0.548 | FT(Warnecke et al., 2023) | 94.267 | 99.998 | 94.403 | 0.107 | 0.553 |
| NGD(Chourasia & Shah, 2023) | 95.133 | 99.654 | 93.280 | 0.081 | 0.544 | NGD(Chourasia & Shah, 2023) | 94.494 | 99.998 | 94.695 | 0.099 | 0.550 |
| NegGrad+(Kurmanji et al., 2023) | 95.578 | 99.731 | 93.300 | 0.082 | 0.549 | NegGrad+(Kurmanji et al., 2023) | 94.115 | 99.998 | 94.173 | 0.113 | 0.565 |
| EUk(Goel et al., 2022) | 99.044 | 99.854 | 93.670 | 0.017 | 0.540 | EUk(Goel et al., 2022) | 98.134 | 99.998 | 92.248 | 0.061 | 0.573 |
| CFk(Goel et al., 2022) | 99.244 | 99.943 | 93.980 | 0.016 | 0.540 | CFk(Goel et al., 2022) | 99.151 | 99.998 | 92.767 | 0.020 | 0.577 |
| SalUn(Fan et al., 2024) | 93.444 | 99.931 | 93.830 | 0.280 | 0.570 | SalUn(Fan et al., 2024) | 92.189 | 98.539 | 91.860 | 0.287 | 0.555 |
| SCRUB(Kurmanji et al., 2023) | 99.222 | 99.511 | 94.060 | 0.047 | 0.548 | SCRUB(Kurmanji et al., 2023) | 99.135 | 99.407 | 92.790 | 0.014 | 0.561 |
| BT(Chundawat et al., 2023) | 91.422 | 99.341 | 93.010 | 0.560 | 0.558 | BT(Chundawat et al., 2023) | 91.703 | 99.287 | 90.300 | 0.633 | 0.608 |
| $l1$-sparse(Jia et al., 2023) | 92.889 | 97.360 | 90.980 | 0.129 | 0.539 | $l1$-sparse(Jia et al., 2023) | 92.098 | 98.020 | 91.165 | 0.140 | 0.548 |

Table 3: Class unlearning of DDPM on CIFAR-10.

Table 4: Image generations of **OPC** for DDPM on CIFAR-10. The forgetting class is 'airplane'.

| Methods | UA (↑) | FID (↓) |
|---|---|---|
| Pretrained | 3.60 | 15.67 |
| Retrain | 99.97 | 13.49 |
| SalUn (Fan et al., 2024) | 99.99 | 17.33 |
| **OPC** (ours) | 99.98 | 16.06 |

| Methods | Forgetting class: 'Airplane' | | | | Non-forgetting classes | | | | | | | | |
|---|---|---|---|---|---|---|---|---|---|---|---|---|---|
| | I1 | I2 | I3 | I4 | C1 | C2 | C3 | C4 | C5 | C6 | C7 | C8 | C9 |
| SalUn | | | | | | | | | | | | | |
| OPC | | | | | | | | | | | | | |

the proposed OPC successfully lowered the forget accuracy even further than retraining without causing significant degradation on $\mathcal{D}_r$. The $\mathbf{MIA}^p$ score is slightly higher for OPC, which may be attributed to its stronger forgetting, but the gap compared to retraining is not considered significant.

## 5 OPC ON DIFFUSION MODELS

The core idea of OPC, collapsing model predictions to a single point (the origin), is not limited to classification models and can be applied to various representation learning settings. As shown in the generative decoder results (Section 4.3.2), minimizing Eq. (1) selectively removes information from forget features, and FM-recovery helps the denoising model generate realistic images from unlearned features.

In this section, we extend OPC to generative models, applying it to the DDPM (Ho et al., 2020) trained on CIFAR10 and the Stable Diffusion (Rombach et al., 2022) model to evaluate its generalizability. Implementation details are provided in Section C.2.

### 5.1 DDPM UNLEARNING

In this section, we aim to unlearn the DDPM model which trained on CIFAR10 to generate image conditioned by class embedding vector, to evaluate naive approach of OPC: push features toward 0 on $\mathcal{D}_f$ and minimize objective loss on $\mathcal{D}_r$

In implementation, we consider the class embedding module of the model as $f_\theta$ and replace the cross entropy loss of Eq. (1) to DDPM loss. In contrast to classification, apply OPC loss to features, as no prediction head is included in the model architecture. The modified loss function can be written as:

$$\mathcal{L}_{OPC}^{DDPM} = \mathbb{E}_{(x_0,c)\sim\mathcal{D}_r,t,\epsilon\sim\mathcal{N}(0,1)}\|\epsilon-\epsilon_\theta(\sqrt{\bar{\alpha}_t}x_0+\sqrt{1-\bar{\alpha}_t}\epsilon, f_\theta(c),t)\|_2^2+\mathbb{E}_{(x_0,c)\sim\mathcal{D}_f}\|f_\theta(c)\|_2 \quad (5)$$

where $c$ represents the class label of image. In experiment, we consider to unlearn single class, the "airplane" whose class label is 0, from the pretrained DDPM.

The results are in Table 3. Consistent to the results on classification model, OPC could guide to unlearn the target class with high UA. Although we pushed the embedding of forget class toward 0, the denoising model could generate high fidelity image from OPC-unlearned class embedding, as FID score of Table 3 remains fine.

### 5.2 STABLE DIFFUSION UNLEARNING

In this section, we aim to unlearn the text-to-image Stable Diffusion (SD) model and evaluate with UnlearnCanvas (Zhang et al., 2024b) benchmark, which requires to unlearn specific styles or object

Table 5: Performance of DM unlearning methods on UnlearnCanvas, measured by UA, IRA, CRA, and FID.

| Method | Effectiveness | | | | | | | | Efficiency | |
|---|---|---|---|---|---|---|---|---|---|---|
| | Style Unlearning | | | Object Unlearning | | | Avg. (↑) | FID (↓) | Memory (GB) (↓) | Storage (GB) (↓) |
| | UA (↑) | IRA (↑) | CRA (↑) | UA (↑) | IRA (↑) | CRA (↑) | | | | |
| ESD (Gandikota et al., 2023) | **98.58%** | 80.97% | 93.96% | 92.15% | 55.78% | 44.23% | 77.61% | 65.55 | 17.8 | 4.3 |
| FMN (Zhang et al., 2024a) | 88.48% | 56.77% | 46.60% | 45.64% | 90.63% | 73.46% | 66.93% | 131.37 | 17.9 | 4.2 |
| UCE (Gandikota et al., 2024) | **98.40%** | 60.22% | 47.71% | 94.31% | 39.35% | 34.67% | 62.45% | 182.01 | 5.1 | 1.7 |
| CA (Kumari et al., 2023) | 60.82% | **96.01%** | 92.70% | 46.67% | 90.11% | 81.97% | 78.05% | 54.21 | 10.1 | 4.2 |
| SalUn (Fan et al., 2024) | 86.26% | 90.39% | **95.08%** | 86.91% | **96.35%** | 99.59% | 92.43% | 61.05 | 30.8 | 4.0 |
| SEOT (Li et al., 2024) | 56.90% | 94.68% | 84.31% | 23.25% | **95.57%** | 82.71% | 72.91% | 62.38 | 7.34 | 0.0 |
| SPM (Lyu et al., 2024) | 60.94% | 92.39% | 84.33% | 71.25% | 90.79% | 81.65% | 80.23% | 59.79 | 6.9 | 0.0 |
| EDiff (Wu et al., 2025) | 92.42% | 73.91% | **98.93%** | 86.67% | 94.03% | 48.48% | 82.41% | 81.42 | 27.8 | 4.0 |
| SHS (Wu & Harandi, 2024) | **95.84%** | 80.42% | 43.27% | 80.73% | 81.15% | 67.99% | 74.90% | 119.34 | 31.2 | 4.0 |
| SAeUron (Cywiński & Deja, 2025) | 95.80% | **99.10%** | **99.40%** | 78.82% | 95.47% | 95.58% | 94.03% | 62.15 | 2.8 | 0.2 |
| **OPC** (ours) | 97.50% | 97.00% | 98.38% | 95.49% | 98.38% | 95.63% | **97.06%** | 55.16 | 9.5 | 0.5 |

while retaining the object or style requirement in prompt, respectively. Instead of updating full diffusion model, whose computation cost is expensive, we aim to edit text encoder $f_\theta$ in perspective of representation learning with low computation cost for training.

Recall Section 4.3.2, the training dynamics of minimizing OPC loss $\mathcal{L}_{OPC}$ (Eq. (1)) could selectively remove the details and FM-recovery layer allows to generate desired images both on forget feature and retain feature. Motivated on this observation, we propose to use auxiliary linear classifier heads $g^{ID}$ and $g^{CD}$ for in-domain classification and cross-domain classification respectively. Those heads would be deleted after the unlearning was performed.

The unlearning process is performed by minimizing $\mathcal{L}_{OPC}$ with in-domain classifier $\mathbf{m}_\theta = g^{ID} \circ f_\theta$ with in-domain class label $y^{ID}$ together with cross-domain $\mathcal{L}_{CE}$ computed on $(g^{CD} \circ f_\theta)(x)$. In particular, the overall loss function can be summarized as:

$$\mathcal{L}_{OPC}^{SD} = \mathcal{L}_{OPC} + \mathbb{E}_{(x,y^{CD}) \sim \mathcal{D}_r \cup \mathcal{D}_f} \mathcal{L}_{CE}((g^{CD} \circ f_\theta)(x), y^{CD}) \tag{6}$$

where $y^{CD}$ is a cross-domain class label. During unlearning, $g^{ID}$ is trainable, while $g^{CD}$ remains frozen.

After getting $\theta^{un}$ by minimizing $\mathcal{L}_{OPC}^{SD}$, we apply FM-recovery explained in Section 4.3 to map $f_{\theta^{un}}(x)$ to pretrained features, to fit the denoising network of diffusion model. Unlike in Section 4.3, where the FM was derived from the validation set, here we construct the recovery layer $W^*$ using only the retain set to avoid mapping information from the forget set. Since FM-recovery layer $W^*$ is linear, this operation may be merged into last layer of $f_\theta$ or the cross attention layer of the denoising network $\epsilon_\theta$.

We follow the instruction of Zhang et al. (2024b) and report the performance of unlearned model in UA, IRA (in-domain retain accuracy), CRA (cross-domain retain accuracy) and FID score. As summarized in Table 5, OPC achieves superior results on both style unlearning and object unlearning, while Zhang et al. (2024b) observed that no single unlearning method consistently excels across all domains, OPC attains over 95% performance in every metric and achieves an average score exceeding 97%, demonstrating robust effectiveness across domains.

Not limited on accuracies, OPC shows superior quality on generated images, with the second-best FID score indicating high fidelity. We show examples of generated images on forget prompt in Fig. 5 and retain prompt in Fig. D.6. OPC-unlearned model successfully generates the desired object in other style (mostly in photo) if unlearning target is style unlearning, and generates only texture without object when the prompt requires to generate the forgotten objective.

## 6    DISCUSSION

A central limitation of prior MU approaches lies in their shallowness. While many methods claim to effectively erase the influence of the forget set, our analyses in Section 4.2 and Section 4.3 reveal that unlearned features remain highly correlated with those of the pretrained model. This residual correlation enables substantial recovery of accuracy on the forget set and even image reconstruction through generative decoders. Such findings indicate that conventional evaluation metrics may over-

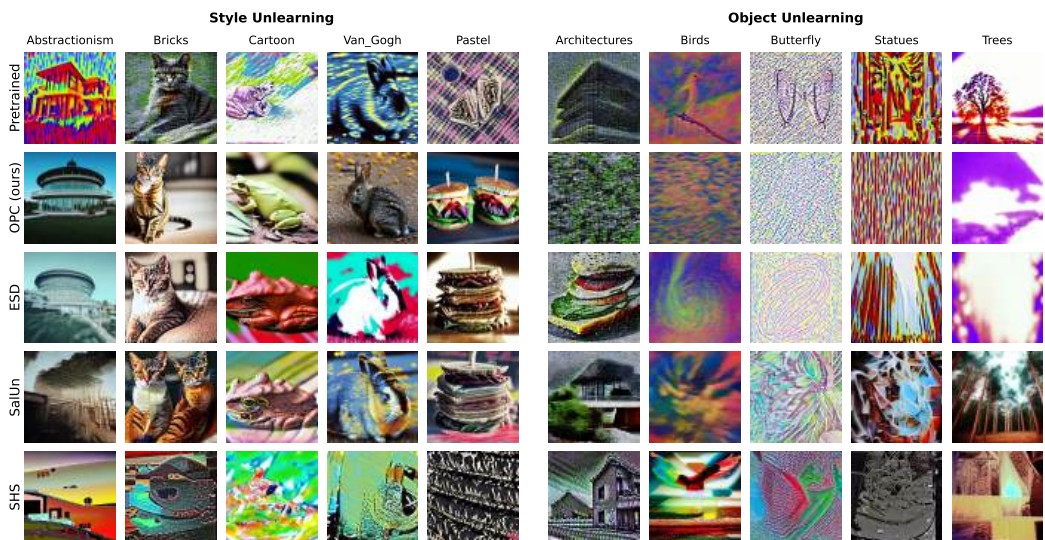

Figure 5: Inference examples of UnlearnCanvas unlearning

state forgetting efficacy, as shallow erasure at the logit level leaves vulnerable traces at the feature level.

In contrast, our proposed method OPC demonstrates strong robustness. Grounded in a clear theoretical framework, OPC enforces contraction of forget-set features within the encoder $f(\cdot)$, thereby erasing informative content rather than merely altering outputs. The empirical evidence confirms that OPC-unlearned representations resist FM-recovery and inversion-based reconstruction attack, establishing its effectiveness in achieving deep feature forgetting.

Finally, we show that the benefits of OPC extend beyond classification tasks. By applying OPC to generative diffusion models, we demonstrate that auxiliary linear layers can guide in-domain forgetting while retaining cross-domain features, enabling selective unlearning. This extension allows for precise control over forgotten attributes, as reflected in Table 5, where OPC uniquely achieves an overall performance of 97%. Importantly, OPC overcomes a key limitation of prior methods: while earlier approaches succeeded in high-frequency (style) unlearning but struggled with low-frequency (object) forgetting, our method successfully handles both, underscoring its generality and versatility across domains.

## 7 CONCLUSION

We critically examine the shallowness of unlearning delivered by existing MU methods, and introduce a novel perspective of "deep feature forgetting". To achieve deep forgetting, we propose One-Point-Contraction (OPC) that contracts the latent feature representation of the forget set data to the origin. Theoretical analysis shows that OPC induces representation-level forgetting, and predicts innate resistance of OPC to adversaries such as recovery attacks and unlearning inversion. Empirical validations highlight the superior performance and resistance of OPC unlearning, and reveals the widespread shallow unlearning phenomena and the limitations of traditional set of unlearning metrics. Moreover, we extend OPC to generative diffusion models, where it enables selective unlearning of style and object attributes. While Zhang et al. (2024b) observed that a single unlearning method can perform differently across various domains and no method excels in all aspects, OPC uniquely achieves over 95% performance in every domain and 97% overall on the UnlearnCanvas benchmark, demonstrating its generality and effectiveness beyond classification.

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

## LLM USAGE

We used LLMs for limited purpose for editing the manuscript only. LLMs were not used for any other purposes.

## A  PROOF OF THEOREM 3.1

**Theorem 3.1.** *Let $C$ be number of classes. Suppose $\mathbf{h} = \mathbf{m}_\theta(x) \in B_r(0)$ where $B_r(0)$ is the ball of radius $r$ centered at origin. Then the entropy $H(softmax(\mathbf{h}))$ of predicted probability has following lower bound parameterized by $r$ and $C$:*

$$H^*(r, C) := \min_{\mathbf{h} \in B_r(0)} H(softmax(\mathbf{h})) > \log\left(1 + (C-1)\exp\left(-\sqrt{\frac{C}{C-1}}r\right)\right) \qquad (2)$$

*Proof.* For the clarity, we denote $\mathbf{q} = \exp(\mathbf{h})$ and $\mathbf{y} = softmax(\mathbf{h}) = \frac{\mathbf{q}}{\|\mathbf{q}\|_1}$.

Let $X = \exp(B_r(0))$ in $\mathbf{q}$-space and $Y = softmax(B_r(0))$ in $\mathbf{y}$-space. Since entropy function $H$ is concave in $\mathbf{y}$-space, the minimal solution $\mathbf{y}^* = argminH(\mathbf{y})$ must lie in the boundary of $Y$, $\partial Y$.

Since $Y$ is a image of $X$ under projection $\mathbf{q} \mapsto \frac{\mathbf{q}}{\|\mathbf{q}\|_1}$ and thus $H(\frac{\mathbf{q}}{\|\mathbf{q}\|_1}) = H(\frac{c\mathbf{q}}{\|c\mathbf{q}\|_1})$ for all $c > 0$, the condition $\mathbf{y}^* = \frac{\mathbf{q}^*}{\|\mathbf{q}^*\|_1} \in \partial Y$ would be translated to followings in $\mathbf{q}$-space:

1. $\mathbf{q}^* \in \partial X$

2. The tangent space $T_{\mathbf{q}^*}X$ includes the origin, $0$.

Since $X = \exp(B_r(0))$, the $\partial X$ would be given by

$$\partial X = \{\mathbf{q} | \sum_{i=1}^{C}(\log q_i)^2 = r^2\} \qquad (A.1)$$

and $T_{\mathbf{q}^*}(X)$ would be

$$T_{\mathbf{q}^*}(X) = \{\mathbf{q} | \sum_{i=1}^{C} \frac{\log q_i^*}{q_i^*}(q_i - q_i^*) = 0\}. \qquad (A.2)$$

Hence, we get $\sum_{i=1}^{C} \log q_i^* = 0$ since $0 \in T_{\mathbf{q}^*}X$.

Therefore, we can find $q^*$ by solving the following constrianed optimization problem.

$$\text{minimize } H(\frac{\mathbf{q}}{\|\mathbf{q}\|_1})$$
$$\text{subject to } \sum_{i=1}^{C} \log q_i = 0 \qquad (A.3)$$
$$\sum_{i=1}^{C}(\log q_i)^2 = r^2$$

Or equivlalently in $\mathbf{h}$-space:

$$\text{minimize } H(softmax(\mathbf{h}))$$
$$\text{subject to } \sum_{i=1}^{C} h_i = 0 \qquad (A.4)$$
$$\sum_{i=1}^{C} h_i^2 = r^2$$

For better readability, we denote $f(\mathbf{h}) = H(softmax(\mathbf{h})) = H(\mathbf{y})$ , $g_1(\mathbf{h}) = \sum_{i=1}^{C} h_i$ and $g_2(\mathbf{h}) = -\frac{r^2}{2} + \sum_{i=1}^{C} \frac{h_i^2}{2}$ and assume $h_1 \geq \cdots \geq h_C$ without loss of generality.

Now let $\lambda_1$ and $\lambda_2$ are the the Lagrangian multipliers, then $\mathbf{h}^*$ should satisfy the stationary condition of Lagrangian, given by $\nabla f(\mathbf{h}) + \lambda_1 \nabla g_1(\mathbf{h}) + \lambda_2 \nabla g_2(\mathbf{h}) = 0$.

Then, by Lemma A.1, we can write $h_1 = \cdots h_b \geq h_{b+1} = \cdots h_C$ for some $b \leq C$ because $h_i$s can have no more than two values.

Now, we can find $h_1$ and $h_C$ from $g_1(\mathbf{h}) = g_2(\mathbf{h})$ for each $b$ that

$$h_1 = \sqrt{\frac{C-b}{bC}}r, h_C = -\sqrt{\frac{b}{C(C-b)}}r \tag{A.5}$$

, which are the stationary points of Lagrangian.

Considering the characteristic of entropy, which is minimized when only one entry is large and rest are small, the optimal $b$ would be $b = 1$. This gives the minimizer

$$\mathbf{h}^* = (\sqrt{\frac{C-1}{C}}r, -\frac{r}{\sqrt{C(C-1)}}, \cdots - \frac{r}{\sqrt{C(C-1)}}). \tag{A.6}$$

Letting $u = -\frac{r}{\sqrt{C(C-1)}}$ and $v = \sqrt{\frac{C}{C-1}}r$, we can rewrite $\mathbf{h}^* = (u+v, u, \cdots, u)$ and obtain

$$\mathbf{y}^* = (\frac{e^v}{e^v + C - 1}, \frac{1}{e^v + C - 1}, \cdots, \frac{1}{e^v + C - 1}). \tag{A.7}$$

Letting $\kappa = \frac{e^v}{C-1}$, the minimal entropy $H(\mathbf{y}^*)$ is given by

$$\begin{aligned}
H(\mathbf{y}^*) &= -\frac{e^v}{e^v + C - 1}(v - \log(e^v + C - 1)) + (C-1)\frac{\log(e^v + C - 1)}{e^v + C - 1} \\
&= \log(e^v + C - 1) - \frac{e^v v}{e^v + C - 1} \\
&= \log((\kappa + 1)(C-1)) - \frac{\kappa(C-1)\log(\kappa(C-1))}{(\kappa+1)(C-1)} \\
&= \log(\kappa+1) + \log(C-1) - \frac{\kappa}{\kappa+1}(\log(\kappa) + \log(C-1)) \\
&= \frac{\log(C-1)}{\kappa+1} + \log(\frac{\kappa+1}{\kappa}) + \frac{\log(\kappa)}{\kappa+1} \\
&= \log(1 + \frac{1}{\kappa}) + \frac{\log(\kappa(C-1))}{\kappa+1}.
\end{aligned} \tag{A.8}$$

Since $\kappa > 0$ and $\log(\kappa(C-1)) = \log(e^v) = \sqrt{\frac{C-1}{C}}r > 0$, we have

$$H(\mathbf{y}^*) > \log(1 + \frac{1}{\kappa}) = \log(1 + (C-1)e^{-v}) = \log(1 + (C-1)\exp(-\sqrt{\frac{C}{C-1}}r)). \tag{A.9}$$

$\square$

**Lemma A.1.** *Let* $f(\mathbf{h}) = H(softmax(\mathbf{h})) = H(\mathbf{y})$ *,* $g_1(\mathbf{h}) = \sum_{i=1}^{C} h_i$ *and* $g_2(\mathbf{h}) = -\frac{r^2}{2} + \sum_{i=1}^{C} \frac{h_i^2}{2}$, *where* $h = (h_1, \cdots, h_C)^T$ *is a variable vector. Suppose that* $\nabla f(h) + \lambda_1 \nabla g_1(h) + \lambda_2 \nabla g_2(h) = 0$. *If* $h_\alpha \geq h_\beta \geq h_\gamma$ *for* $\alpha, \beta, \gamma \in [C]$ *then at least two of them must be equal. i.e.* $h_\alpha = h_\beta$ *or* $h_\beta = h_\gamma$.

*Proof.* Consider $3 \times C$ matrix $M$, whose row vectors are $\nabla g_1$, $\frac{1}{2}\nabla g_2$ and $\nabla f$. and its submatrix $M_{\alpha,\beta,\gamma}$ consist of $\alpha, \beta, \gamma$=th entries. By simple differentiation, it would be

$$M_{\alpha,\beta,\gamma} = \begin{bmatrix} 1 & 1 & 1 \\ h_\alpha & h_\beta & h_\gamma \\ \frac{\partial}{\partial h_\alpha}H(\mathbf{y}) & \frac{\partial}{\partial h_\beta}H(\mathbf{y}) & \frac{\partial}{\partial h_\gamma}H(\mathbf{y}) \end{bmatrix} \tag{A.10}$$

Since $rank M \leq 2$ by assumption, $rank M_{\alpha,\beta,\gamma} \leq 2$ and thus we can find $c_\alpha, c_\beta, c_\gamma$ who are not all zero, satisfying

$$
\begin{aligned}
& c_\alpha + c_\beta + c_\gamma = 0 \\
& c_\alpha h_\alpha + c_\beta h_\beta + c_\gamma h_\gamma = 0 \\
& c_\alpha \frac{\partial}{\partial h_\alpha} H(\mathbf{y}) + c_\beta \frac{\partial}{\partial h_\beta} H(\mathbf{y}) + c_\gamma \frac{\partial}{\partial h_\gamma} H(\mathbf{y}) = 0
\end{aligned}
\tag{A.11}
$$

If $c_\beta = 0$, then $c_\alpha = -c_\gamma$ and thus $h_\alpha = h_\beta = h_\gamma$. otherwise, letting $\delta = -\frac{c_\alpha}{c_\beta}$ then we have $h_\beta = \delta h_\alpha + (1-\delta)h_\gamma$ and $\delta \in [0,1]$ since $h_\alpha \geq h_\beta \geq h_\gamma$.

Since $e^x$ is convex, we have $\delta e^{h_\alpha} + (1-\delta)e^{h_\gamma} \geq e^{h_\beta}$ and $S := \delta y_\alpha + (1-\delta)y_\gamma \geq y_\beta$ because $y_i = \frac{e^{h_i}}{\sum_{j=1}^C e^{h_j}}$.

Now we compute the $\frac{\partial}{\partial h_i} H(\mathbf{y})$. From the chain rule, we have

$$
\frac{\partial}{\partial h_i} H(\mathbf{y}) = \sum_{k=1}^C \frac{\partial y_k}{\partial h_i} \frac{\partial H(\mathbf{y})}{\partial y_k}.
\tag{A.12}
$$

From simple computation, $\frac{\partial H(\mathbf{y})}{\partial y_k} = -(1 + \log(y_k))$ and

$$
\frac{\partial y_k}{\partial h_i} = \begin{cases}
-\frac{e^{h_i} e^{h_k}}{(\sum_{j=1}^C e^{h_j})^2} = -y_i y_k & \text{if } i \neq k \\
\frac{e^{h_i}}{\sum_{j=1}^C e^{h_j}} - \frac{e^{2h_i}}{(\sum_{j=1}^C e^{h_j})^2} = y_i - y_i^2 & \text{if } i = k
\end{cases}
\tag{A.13}
$$

Therefore, we can summarize

$$
\begin{aligned}
\frac{\partial}{\partial h_i} H(\mathbf{y}) &= -y_i(1 + \log(y_i)) + \sum_{k=1}^C y_i y_k (1 + \log(y_k)) \\
&= -y_i \log(y_i) - y_i(H(\mathbf{y})) = -y_i(\log(y_i) + H(\mathbf{y})).
\end{aligned}
\tag{A.14}
$$

The third equation of Eq. (A.11) is now written as

$$
\delta y_\alpha(\log(y_\alpha) + H) + (1-\delta)y_\gamma(\log(y_\gamma) + H) = y_\beta(\log(y_\beta) + H)
\tag{A.15}
$$

were $H(\mathbf{y})$ is simplified to $H$.

Now we suppose $y_\alpha \neq y_\gamma$ and $\delta y_\alpha \log(y_\alpha) + (1-\delta)y_\gamma \log(y_\gamma) < y_\beta \log(y_\beta)$.

Recall the $S = \delta y_\alpha + (1-\delta)y_\gamma \geq y_\beta$ and $\log(y_\beta) = \delta \log(y_\alpha) + (1-\delta)\log(y_\gamma)$, we have

$$
\delta y_\alpha \log(y_\alpha) + (1-\delta)y_\gamma \log(y_\gamma) < y_\beta \log(y_\beta) \leq S \log(y_\beta) = \delta S \log(y_\alpha) + (1-\delta)S \log(y_\gamma)
\tag{A.16}
$$

and thus

$$
\delta(1-\delta)(y_\alpha - y_\gamma)\log(y_\alpha) = \delta(y_\alpha - S)\log(y_\alpha) < (1-\delta)(S - y_\gamma)\log(y_\gamma) = \delta(1-\delta)(y_\alpha - y_\gamma)\log(y_\gamma).
\tag{A.17}
$$

This concludes that $\log(y_\alpha) < \log(y_\gamma)$ because $\delta > 0, 1 - \delta > 0$ and $(y_\alpha - y_\gamma) > 0$, which is contradiction because $h_\alpha \geq h_\gamma$. Hence, $y_\alpha = y_\gamma$ or $\delta y_\alpha \log(y_\alpha) + (1-\delta)y_\gamma \log(y_\gamma) \geq y_\beta \log(y_\beta)$.

If $y_\alpha = y_\gamma$ then proof is finished. Otherwise, from $H > 0$ and $\delta y_\alpha + (1-\delta)y_\gamma \geq y_\beta$ we can obtain the inequality

$$
\delta y_\alpha(\log(y_\alpha) + H) + (1-\delta)y_\gamma(\log(y_\gamma) + H) \geq y_\beta(\log(y_\beta) + H)
\tag{A.18}
$$

where equality holds iff $\delta = 0$ or $\delta = 1$. Since we have Eq. (A.15), we conclude $\delta = 0$ or $\delta = 1$, and finally $h_\gamma = h_\beta$ or $h_\alpha = h_\beta$.

$\square$

## B  UNLEARNING ALGORITHMS

Gradient Ascent (GA) attempts to undo learning from retain set by reversing gradient directions Thudi et al. (2022). Random Labeling (RL) trains the model using retain set and randomly labeled forget set Golatkar et al. (2020). Boundary Expanding (BE) pushes forget set to an extra shadow class Chen et al. (2023). Fine Tuning (FT) continues training on retain set using standard stochastic gradient descent (SGD) Warnecke et al. (2023). Noisy Gradient Descent (NGD) modifies FT by adding Gaussian noise to each update step Chourasia & Shah (2023). Exact Unlearning the last k layers (EUk) retrains only the last k layers from scratch to remove forget set information. Catastrophically Forgetting the last k layers (CFk), instead of retraining, continues training the last k layers on retain set Goel et al. (2022). Saliency Unlearning (SalUn) enhances RL by freezing important model weights using gradient-based saliency maps Fan et al. (2024). Bad-Teacher (BT) uses a student-teacher framework where the teacher is trained on full train set and the student mimics it for retain set, while imitating a randomly initialized model, the "bad teacher", for forget set Chundawat et al. (2023). SCalable Remembering and Unlearning unBound (SCRUB), a state-of-the-art technique, also employs a student-teacher setup to facilitate unlearning. NegGrad+ combines GA and FT to fine-tune the model in a way that effectively removes forget set information Kurmanji et al. (2023). $l1$-sparse enhances FT with $l1$ regularization term Jia et al. (2023). Selective Synaptic Dampening (SSD) unlearns by dampening weights that strongly influence the Fisher information of the forget set more than the rest of the dataset Foster et al. (2024).

For diffusion model unlearning, EDiff (Wu et al., 2025) formulates the task as a bi-level optimization problem, ESD (Gandikota et al., 2023) adopts negative classifier-free guidance, and FMN (Zhang et al., 2024a) proposes a re-steering loss applied only to attention layers. SalUn (Fan et al., 2024) and SHS (Wu & Harandi, 2024) adapt parameters based on saliency maps or connection sensitivity, while SA (Heng & Soh, 2023) replaces the original distribution of unwanted data with a surrogate one, extended to anchor concepts in CA (Kumari et al., 2023). SPM (Lyu et al., 2024) takes another route by introducing small linear adapters after each linear and convolutional layer to block the propagation of undesired information.

In contrast, non–fine-tuning approaches include SEOT (Li et al., 2024), which removes unwanted content directly from text embeddings, and UCE (Gandikota et al., 2024), which modifies cross-attention weights through a closed-form solution. Distinct from these, SAeUron (Cywiński & Deja, 2025) leverages sparse autoencoders (SAEs) to effectively eliminate undesired concepts in text-to-image diffusion models.

## C  EXPERIMENTAL SETUP DETAILS

### C.1  CLASSIFICATION MODELS

In this section, we detail the experimental settings in Section 4.1. All experiments were conducted on a machine equipped with an AMD Ryzen 9 5900X 12-Core CPU, an NVIDIA GeForce RTX 3090 GPU with 24GB of VRAM, and 64GB of TEAMGROUP UD4-3200 RAM (2 × 32GB). To obtain the pretrained models, we trained ResNet-18 (He et al., 2016) from scratch on CIFAR-10 (Krizhevsky et al., 2010) and SVHN (Netzer et al., 2011) datasets. The pretrained model was trained for 182 epochs with a learning rate of 0.1 on CIFAR-10, and for 200 epochs with a learning rate of

Table C.1: Table of training information on 30% Class unlearning scenario

| CIFAR10 | Epochs | Learning rate | Runtime (s) | SVHN | Epochs | Learning rate | Runtime (s) |
|---|---|---|---|---|---|---|---|
| Retrain | 182 | 0.01 | 3,547.403 | Retrain | 182 | 0.01 | 4,185.296 |
| OPC (ours) | 30 | 0.01 | 1,019.318 | OPC (ours) | 25 | 0.01 | 1,152.792 |
| GA(Thudi et al., 2022) | 10 | 0.00004 | 86.469 | GA(Thudi et al., 2022) | 5 | 0.000005 | 76.621 |
| RL(Golatkar et al., 2020) | 15 | 0.018 | 424.281 | RL(Golatkar et al., 2020) | 15 | 0.013 | 547.849 |
| BE(Chen et al., 2023) | 10 | 0.0001 | 87.335 | BE(Chen et al., 2023) | 4 | 0.0000185 | 58.914 |
| FT(Warnecke et al., 2023) | 20 | 0.035 | 394.531 | FT(Warnecke et al., 2023) | 20 | 0.035 | 450.431 |
| NGD(Chourasia & Shah, 2023) | 20 | 0.035 | 401.088 | NGD(Chourasia & Shah, 2023) | 20 | 0.035 | 440.530 |
| NegGrad+(Kurmanji et al., 2023) | 20 | 0.035 | 656.626 | NegGrad+(Kurmanji et al., 2023) | 15 | 0.035 | 565.179 |
| EUk(Goel et al., 2022) | 20 | 0.035 | 289.609 | EUk(Goel et al., 2022) | 20 | 0.035 | 298.624 |
| CFk(Goel et al., 2022) | 20 | 0.04 | 281.858 | CFk(Goel et al., 2022) | 40 | 0.1 | 578.894 |
| SalUn(Fan et al., 2024) | 20 | 0.02 | 288.443 | SalUn(Fan et al., 2024) | 15 | 0.015 | 250.583 |
| SCRUB(Kurmanji et al., 2023) | 3 | 0.0003 | 84.362 | SCRUB(Kurmanji et al., 2023) | 15 | 0.00007 | 580.143 |
| BT(Chundawat et al., 2023) | 5 | 0.01 | 589.062 | BT(Chundawat et al., 2023) | 8 | 0.01 | 1,366.039 |
| $l1$-sparse(Jia et al., 2023) | 20 | 0.005 | 397.200 | $l1$-sparse(Jia et al., 2023) | 20 | 0.015 | 455.502 |

Table C.2: Table of training information on 10% random unlearning scenario

| CIFAR10 | Epochs | Learning rate | Runtime (s) | SVHN | Epochs | Learning rate | Runtime (s) |
|---|---|---|---|---|---|---|---|
| Retrain | 182 | 0.01 | 4,648.831 | Retrain | 182 | 0.01 | 5,962.928 |
| OPC (ours) | 20 | 0.009 | 610.043 | OPC (ours) | 5 | 0.0008 | 197.374 |
| GA(Thudi et al., 2022) | 15 | 0.0001 | 41.759 | GA(Thudi et al., 2022) | 15 | 0.0001 | 61.970 |
| RL(Golatkar et al., 2020) | 20 | 0.008 | 560.755 | RL(Golatkar et al., 2020) | 15 | 0.013 | 553.956 |
| BE(Chen et al., 2023) | 8 | 0.00001 | 26.061 | BE(Chen et al., 2023) | 4 | 0.000008 | 15.911 |
| FT(Warnecke et al., 2023) | 40 | 0.1 | 1,016.424 | FT(Warnecke et al., 2023) | 42 | 0.1 | 1,399.713 |
| NGD(Chourasia & Shah, 2023) | 40 | 0.1 | 1,032.924 | NGD(Chourasia & Shah, 2023) | 40 | 0.1 | 1,329.540 |
| NegGrad+(Kurmanji et al., 2023) | 40 | 0.05 | 1,617.294 | NegGrad+(Kurmanji et al., 2023) | 10 | 0.03 | 545.281 |
| EUk(Goel et al., 2022) | 40 | 0.1 | 721.451 | EUk(Goel et al., 2022) | 10 | 0.03 | 220.091 |
| CFk(Goel et al., 2022) | 40 | 0.1 | 719.283 | CFk(Goel et al., 2022) | 10 | 0.03 | 221.769 |
| SalUn(Fan et al., 2024) | 20 | 0.01 | 316.121 | SalUn(Fan et al., 2024) | 15 | 0.01 | 275.977 |
| SCRUB(Kurmanji et al., 2023) | 3 | 0.002 | 84.950 | SCRUB(Kurmanji et al., 2023) | 5 | 0.000038 | 193.303 |
| BT(Chundawat et al., 2023) | 12 | 0.01 | 1,442.486 | BT(Chundawat et al., 2023) | 2 | 0.005 | 337.738 |
| $l1$-sparse (Jia et al., 2023) | 25 | 0.01 | 643.387 | $l1$-sparse (Jia et al., 2023) | 20 | 0.01 | 670.176 |

Table C.3: Table of hyperparameters on unlearning scenario

| Methods | Hparam name | Description of hyperparameters | 30% Class | 10% random |
|---|---|---|---|---|
| OPC(Ours) | $coeff\_ce$ | weight for the cross-entropy loss on retain data, | 1 | 0.95 |
| | $coeff\_un$ | weight for the norm loss on forget data | 0.7 | CIFAR10:0.05, SVHN:0.2 |
| NGD(Chourasia & Shah, 2023) | $\sigma$ | standard deviation of Gaussian noise added to gradients | $10^{-7}$ | $10^{-7}$ |
| NegGrad+(Kurmanji et al., 2023) | $\alpha$ | controls weighted mean of retain and forget losses | 0.999 | 0.999 |
| EUk(Goel et al., 2022) | $k$ | Last $k$ layers to be trained | 3 | 3 |
| CFk(Goel et al., 2022) | $k$ | Last $k$ layers to be trained | 3 | 3 |
| SalUn(Fan et al., 2024) | $pt$ | sparsity ratio for weight saliency | 0.5 | 0.5 |
| SCRUB(Kurmanji et al., 2023) | $\alpha$ | weight of KL loss between student and teacher. | 0.001 | 0.001 |
| | $\beta$ | scales optional extra distillation loss | 0 | 0 |
| | $\gamma$ | weight of classification loss. | 0.99 | 0.99 |
| | $kd\_T$ | controls the softening of softmax outputs for distillation. | 4 | 4 |
| | $msteps$ | # of maximize steps using forget data before minimize training. | CIFAR10:2, SVHN:1 | 1 |
| $l1$-sparse (Jia et al., 2023) | $\alpha$ | weight of $l1$ regularization | 0.0001 | 0.0001 |

0.1 on SVHN. The optimizer used in our experiments was Stochastic Gradient Descent (SGD) with a momentum of 0.9 and a weight decay of 1e-5. For learning rate scheduling, we employed PyTorch's MultiStepLR with milestones set at epochs 91 and 136, and a gamma value of 0.1.

For data augmentation, we applied common settings cosist of RandomCrop(32, 4) and RandomHorizontalFlip, from the torchvision (maintainers & contributors, 2016) library to CIFAR-10 (maintainers & contributors, 2016). No augmentation was used for SVHN, considering its digit-centric nature and the presence of multiple digits in a single image, with only the center digit serving as the target. Unless otherwise stated, we used a batch size of 256 for all training procedures, including pretraining.

The training epochs and learning rates used for each unlearning method in Section 4.1 are listed in Table C.1 and Table C.2. Based on these settings, the runtime of each method can also be checked. On Class unlearning scenario, **OPC** generally takes longer to run. This is because, while most other methods show degradation of accuracy on $\mathcal{D}_r$ and the test set $test\ \mathcal{D}_r$ as training epochs increase, **OPC** shows improved accuracy with more training.

Other hyperparameters and their descriptions are provided in Table C.3.

## C.2 DIFFUSION MODELS

For DDPM decoder, The model structure and training settings followed Heng & Soh (2023), with two modifications: the addition of a hidden dimension to accept $f_{\theta^0}(x)$ as a conditioning vector, and an increased training budget of 1.26 million iterations.

For DDPM unlearning, we used the hardware described in Section C.1. The architecture, generation of pretrained and retrained DDPM checkpoints, and data preprocessing were implemented following Fan et al. (2024). The evaluation code was also based on Fan et al. (2024), except for the FID score, which followed the implementation of Seitzer (2020). Training was performed with a batch size of 64 over 40,000 iterations, with the hyperparameter $coeff\_un$ set to 0.2, as specified in Table C.3.

For SD unlearning, experiments were carried out on an NVIDIA A100 80GB GPU. Only text data (a total of 1,020 samples) was used, trained with a batch size of 64 for 1,000 epochs. In cases such as *Human* and *Trees*, where unlearning appeared less effective, training was extended to 2,000 epochs.

Table D.1: Unlearning performance with train-free unlearning on prediction head only

| CIFAR10 | Train $\mathcal{D}_f$ | Train $\mathcal{D}_r$ | test $\mathcal{D}_f$ | Test $\mathcal{D}_r$ | MIA$^e$ | SVHN | Train $\mathcal{D}_f$ | Train $\mathcal{D}_r$ | test $\mathcal{D}_f$ | Test $\mathcal{D}_r$ | MIA$^e$ |
|---|---|---|---|---|---|---|---|---|---|---|---|
| Pretrained | 99.444 | 99.416 | 94.800 | 94.400 | 0.015 | Pretrained | 99.531 | 99.172 | 94.960 | 91.110 | 0.009 |
| Retrain | 0.000 | 99.981 | 0.000 | 91.700 | 1.000 | Retrain | 0.000 | 99.997 | 0.000 | 92.440 | 1.000 |
| OPC-TF | 0.363 | 99.552 | 0.367 | 95.329 | 1.000 | OPC-TF | 0.019 | 99.369 | 0.018 | 92.926 | 1.000 |
| RL-TF | 4.785 | 99.552 | 3.933 | 95.314 | 1.000 | RL-TF | 1.278 | 99.347 | 0.946 | 92.959 | 1.000 |

The learning rate was set to $1e-5$, and optimization was performed using AdamW with parameters $\beta_1 = 0.9, \beta_2 = 0.999$, weight decay of $1e-2$, and epsilon of $1e-8$.

To construct the pretrained auxiliary layer, we trained with a batch size of 64 using cross-entropy loss under the same optimizer configuration as above. Training was conducted for 400 epochs, with the objective of achieving 100% accuracy in both cases.

Finally, the UnlearnCanvas benchmark model checkpoints were obtained by following the directions provided in Zhang et al. (2024b).

# D DETAILED EXPERIMENTAL RESULTS

In this section, we list the detailed results of classifiaction model unleaning on CIFAR10 and SVHN, and diffusion model on UnlearnCanvas which were omitted in Section 4 due to page limit.

## D.1 HEAD RECOVERY OF UNLEARNED MODELS

Previous evaluation in Section 4.3 shows the existence of proper classifier head which allows the recovery of model performance on $\mathcal{D}_f$, but with the oracle of pretrained model. In this section, we aim to try the same without the pretrained model, by mapping the unlearned features directly to the desired logits (the one-hot vector of target labels) with similar method.

We consider following linear least square problem to find the recovered prediction head:

$$W^* = \arg\min_W \sum_{(x,y)\in\mathcal{D}} \|W f_{\theta^{un}}(x) - e_y\|_2^2, \tag{D.1}$$

where $\mathcal{D}$ is a sample dataset, $\theta^{un}$ is the unlearned parameters and $e_y$ is the one-hot vector of label $y$ of sample $x$. We used $\mathcal{D}_{val}$ as sample dataset in implementation. For CIFAR10, we used normalized features instead of $f_{\theta^{un}}(x)$ since some models including retrained model lost performance on $\mathcal{D}_r$.

## D.2 TRAINING-FREE UNLEARNING

In Section 4.4, we showed that class unlearning can be achieved successfully even with minimal forgetting at the feature level. Building on this and Section D.1, we further investigate whether class unlearning can be performed in a train-free manner.

We hypothesize that we can make unlearned model by applying modification only on the prediction head with similar approach, and achieve good performance on logit-based metrics, which are the most common criteria for the MU.

In this section, we solve the least squares problem $\arg\min_W \sum_{x\in\mathcal{D}_f\cup\mathcal{D}_r} \|Wx - \hat{y}\|_2^2$ where $\hat{y} = 0$ if $x \in \mathcal{D}_f$ and otherwise the one-hot vector of true label $\hat{y} = e_{label}$. For the comparison, we also solve least square problem with RL, by providing $\hat{y}$ as the one-hot vector of random label for the forget sample $x \in \mathcal{D}_f$.

The results are in Table D.1. The training-free unlearned prediction head shows near-zero accuracy on $\mathcal{D}_f$, and even better accuracy on $\mathcal{D}_r$ compared to the pretrained model. The training-free head-only unlearning with RL method also shows promising results, but the forgetting was insufficient.

### D.3 UNLEARNING INVERSION ATTACK

Recently, Hu et al. (2024) claimed the vulnerability of MU, with unlearning inversion attack, based on gradient-inversion, on unlearned model. Surprisingly, the attacker could reconstruct the sample image which were in the forget set $\mathcal{D}_f$. To visualize how the unlearning methods forget features, we exploit Hu et al. (2024)'s method and applied it to MU benchmarks and our method, to evaluate the vulnerability under unlearning inversion attack.

Given sample image and corresponding label $(x, y) \in \mathcal{D}_f$ in forget set, the original Hu et al. (2024) implementation takes $\nabla^*$ as the parameter movement driven by unlearning process with single forget sample and find best sample $x'$ which makes $\nabla'(x') = \nabla_\theta \mathcal{L}_{CE}(f_\theta(x'), y)$ similar to $\nabla^*$, but unfortunately the unlearning problem setting does not meet theirs, since the forget set $\mathcal{D}_f$ is much larger compared to the single datapoint used in Hu et al. (2024). Hence, we introduce an oracle providing true $\nabla_\theta \mathcal{L}_{CE}(f_\theta(x), y)$ as $\nabla^*$ for the reconstruction, which is quite strong advantage for the attacker and highly informative.

### D.4 CLASS UNLEARNING

#### D.4.1 UNLEARNING INVERSION ATTACK

We provide more examples of the recovered images from the unlearning inversion attack against the unlearned models on class unlearning scenario.

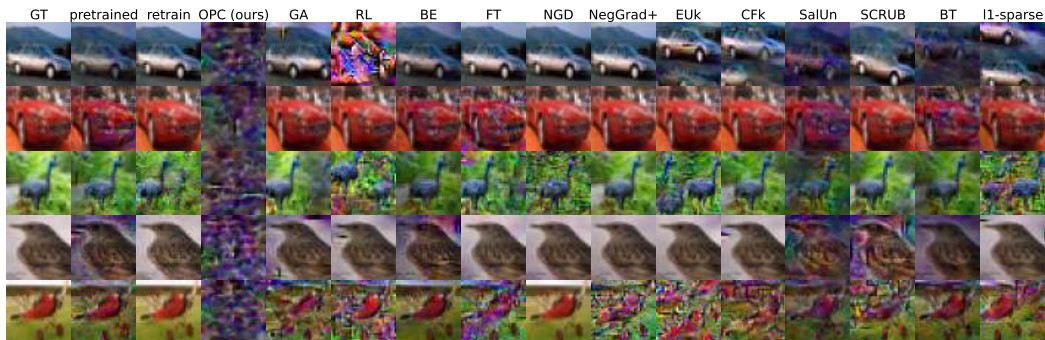

(a) Reconstruction of forgotten images on CIFAR10 30% class unlearning scenario

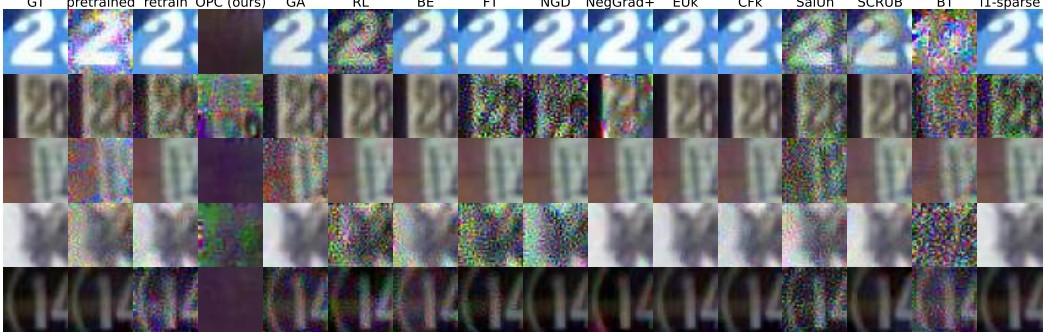

(b) Reconstruction of forgotten images on SVHN 30% class unlearning scenario

Figure D.1: The results of unlearning inversion. The target images are sampled from the forget set $\mathcal{D}_f$ under 30% class unlearning scenario. GT represents the ground truth image from the dataset and others are the results of inversion attacks from each unlearned model.

The results are collected in Fig. D.1. Interestingly, almost all other unlearning methods including retrain were vulnerable under the inversion attack, while only our method **OPC** was consistently resistant. Possibly, this observation would support the loss of discriminative ability of unlearned model induced by our one-point contraction method.

Table D.2: Recovered performance with $W^*$ and pretrained head on 30% Class unlearning scenario

| CIFAR10 | Train $\mathcal{D}_f$ | Train $\mathcal{D}_r$ | test $\mathcal{D}_f$ | test $\mathcal{D}_r$ | $\mathbf{MIA}^e$ | SVHN | Train $\mathcal{D}_f$ | Train $\mathcal{D}_r$ | test $\mathcal{D}_f$ | test $\mathcal{D}_r$ | $\mathbf{MIA}^e$ |
|---|---|---|---|---|---|---|---|---|---|---|---|
| Pretrained | 99.444 | 99.416 | 94.800 | 94.400 | 0.015 | Pretrained | 99.531 | 99.172 | 94.960 | 91.110 | 0.009 |
| Retrain | 70.341 | 95.435 | 70.400 | 86.700 | 0.556 | Retrain | 88.434 | 96.682 | 88.428 | 87.660 | 0.196 |
| OPC (ours) | 45.000 | 99.000 | 44.200 | 90.929 | 0.944 | OPC (ours) | 51.304 | 99.068 | 50.637 | 90.818 | 1.000 |
| GA(Thudi et al., 2022) | 86.622 | 96.010 | 81.733 | 90.500 | 0.283 | GA(Thudi et al., 2022) | 99.422 | 99.161 | 93.959 | 91.237 | 0.014 |
| RL(Golatkar et al., 2020) | 94.356 | 98.711 | 89.233 | 92.086 | 0.121 | RL(Golatkar et al., 2020) | 92.229 | 97.340 | 91.003 | 90.625 | 0.132 |
| BE(Chen et al., 2023) | 99.400 | 99.413 | 94.533 | 93.857 | 0.022 | BE(Chen et al., 2023) | 99.369 | 99.073 | 93.313 | 89.535 | 0.024 |
| FT(Warnecke et al., 2023) | 90.644 | 98.390 | 87.800 | 92.186 | 0.235 | FT(Warnecke et al., 2023) | 94.769 | 98.278 | 93.777 | 91.150 | 0.100 |
| NGD(Chourasia & Shah, 2023) | 89.778 | 98.181 | 85.867 | 92.386 | 0.255 | NGD(Chourasia & Shah, 2023) | 94.111 | 97.862 | 93.577 | 91.789 | 0.110 |
| NegGrad+(Kurmanji et al., 2023) | 87.526 | 97.730 | 84.467 | 91.014 | 0.298 | NegGrad+(Kurmanji et al., 2023) | 94.145 | 96.312 | 93.987 | 91.430 | 0.093 |
| EUk(Goel et al., 2022) | 96.444 | 99.311 | 90.100 | 93.586 | 0.182 | EUk(Goel et al., 2022) | 96.035 | 98.891 | 93.049 | 90.193 | 0.091 |
| CFk(Goel et al., 2022) | 98.711 | 99.613 | 93.000 | 94.386 | 0.080 | CFk(Goel et al., 2022) | 99.210 | 99.661 | 94.141 | 90.605 | 0.034 |
| SalUn(Fan et al., 2024) | 96.081 | 99.432 | 91.333 | 93.314 | 0.092 | SalUn(Fan et al., 2024) | 92.482 | 97.292 | 91.257 | 90.658 | 0.125 |
| SCRUB(Kurmanji et al., 2023) | 89.444 | 97.651 | 84.633 | 92.257 | 0.255 | SCRUB(Kurmanji et al., 2023) | 91.620 | 89.937 | 90.857 | 85.020 | 0.126 |
| BT(Chundawat et al., 2023) | 99.304 | 99.438 | 93.133 | 94.329 | 0.041 | BT(Chundawat et al., 2023) | 94.795 | 98.171 | 92.986 | 89.907 | 0.109 |
| $l1$-sparse(Jia et al., 2023) | 95.274 | 99.184 | 89.900 | 93.343 | 0.169 | $l1$-sparse(Jia et al., 2023) | 92.701 | 96.244 | 91.985 | 89.740 | 0.127 |

Table D.3: Recovered performance with head recovery on 30% Class unlearning scenario

| CIFAR10 | Train $\mathcal{D}_f$ | Train $\mathcal{D}_r$ | test $\mathcal{D}_f$ | test $\mathcal{D}_r$ | $\mathbf{MIA}^e$ | SVHN | Train $\mathcal{D}_f$ | Train $\mathcal{D}_r$ | test $\mathcal{D}_f$ | test $\mathcal{D}_r$ | $\mathbf{MIA}^e$ |
|---|---|---|---|---|---|---|---|---|---|---|---|
| Pretrained | 99.607 | 99.571 | 95.067 | 94.114 | 0.082 | Pretrained | 99.675 | 99.255 | 95.506 | 90.598 | 0.086 |
| Retrain | 71.963 | 95.213 | 72.400 | 85.557 | 0.750 | Retrain | 89.292 | 96.221 | 89.465 | 85.326 | 0.440 |
| OPC (ours) | 33.333 | 99.156 | 31.633 | 91.214 | 0.976 | OPC (ours) | 47.154 | 99.521 | 45.524 | 91.376 | 1.000 |
| GA(Thudi et al., 2022) | 87.096 | 95.305 | 82.400 | 89.871 | 0.413 | GA(Thudi et al., 2022) | 99.572 | 99.124 | 94.733 | 90.386 | 0.129 |
| RL(Golatkar et al., 2020) | 94.207 | 98.679 | 89.333 | 92.071 | 0.246 | RL(Golatkar et al., 2020) | 92.153 | 97.627 | 90.775 | 90.386 | 0.353 |
| BE(Chen et al., 2023) | 99.607 | 99.444 | 94.600 | 93.429 | 0.099 | BE(Chen et al., 2023) | 98.851 | 98.825 | 94.041 | 87.666 | 0.230 |
| FT(Warnecke et al., 2023) | 90.556 | 98.270 | 87.933 | 91.686 | 0.427 | FT(Warnecke et al., 2023) | 94.803 | 98.065 | 94.241 | 90.339 | 0.339 |
| NGD(Chourasia & Shah, 2023) | 89.881 | 98.013 | 87.067 | 92.043 | 0.444 | NGD(Chourasia & Shah, 2023) | 94.606 | 97.604 | 94.023 | 90.412 | 0.351 |
| NegGrad+(Kurmanji et al., 2023) | 86.889 | 97.559 | 84.667 | 90.700 | 0.538 | NegGrad+(Kurmanji et al., 2023) | 93.877 | 96.254 | 93.559 | 90.765 | 0.350 |
| EUk(Goel et al., 2022) | 96.830 | 99.422 | 91.333 | 93.100 | 0.454 | EUk(Goel et al., 2022) | 95.808 | 98.376 | 93.604 | 88.883 | 0.376 |
| CFk(Goel et al., 2022) | 98.644 | 99.800 | 92.867 | 93.829 | 0.292 | CFk(Goel et al., 2022) | 98.632 | 99.321 | 94.778 | 89.834 | 0.264 |
| SalUn(Fan et al., 2024) | 95.956 | 99.406 | 91.500 | 93.200 | 0.208 | SalUn(Fan et al., 2024) | 92.338 | 97.432 | 91.366 | 90.472 | 0.353 |
| SCRUB(Kurmanji et al., 2023) | 88.956 | 97.048 | 84.367 | 91.457 | 0.453 | SCRUB(Kurmanji et al., 2023) | 91.786 | 87.612 | 91.012 | 83.019 | 0.786 |
| BT(Chundawat et al., 2023) | 99.481 | 99.495 | 93.500 | 94.029 | 0.175 | BT(Chundawat et al., 2023) | 93.661 | 98.098 | 92.394 | 89.408 | 0.420 |
| $l1$-sparse(Jia et al., 2023) | 94.963 | 99.149 | 89.667 | 92.671 | 0.372 | $l1$-sparse(Jia et al., 2023) | 92.788 | 95.631 | 92.213 | 88.464 | 0.444 |

### D.4.2 RECOVERY ATTACK RESULTS

We provide the results of recovery attack, including the retain accuracy, test accuracy and $\mathbf{MIA}^e$, in Table D.2. And, the results of head recovery attack in Table D.3. The recovery succeeded to reduce the forget accuracy as shown in Fig. 3 by decrease of UA, while the performance on retain classes are preserved.

### D.4.3 CKA SIMILARITY

In Fig. D.2 we provide the CKA similarity of unlearned models compared to the pretrained model, evaluated on SVHN. Note that CIFAR10 result can be found in Section 4.2.

Similar to CIFAR10 forgetting, **OPC** shows similar results: the near-zero simiarity on the forget dataset and high similarity on retain set. Unlike CIFAR10 results, most of benchmark models are showing lower CKA similarity scores on forget dataset $\mathcal{D}_f$, but not significantly less than **OPC**.

### D.5 RANDOM UNLEARNING

### D.5.1 UNLEARNING INVERSION ATTACK

We provide the recovered images from the unlearning inversion attack against the unlearned models on random unlearning scenario.

Fig. D.3 shows the results. While almost all models show the vulnerability, the **OPC**-unlearned model shows the resistance.

Some forget images were recovered in CIFAR10, but this observation is may due to the imperfect unlearning, since the forget accuracy is still high (but much less than others) in Table 2. The results on SVHN shows the high resistance of **OPC**, as the forgetting was extremely successful with significant gap on forget accuracy (7.5% on **OPC**, > 90 on others).

### D.5.2 CKA SIMILARITY

We measure the CKA similarity of features of unlearned model, compared to the pretrained model, under random unlearning scenario and visualize in Fig. D.4.

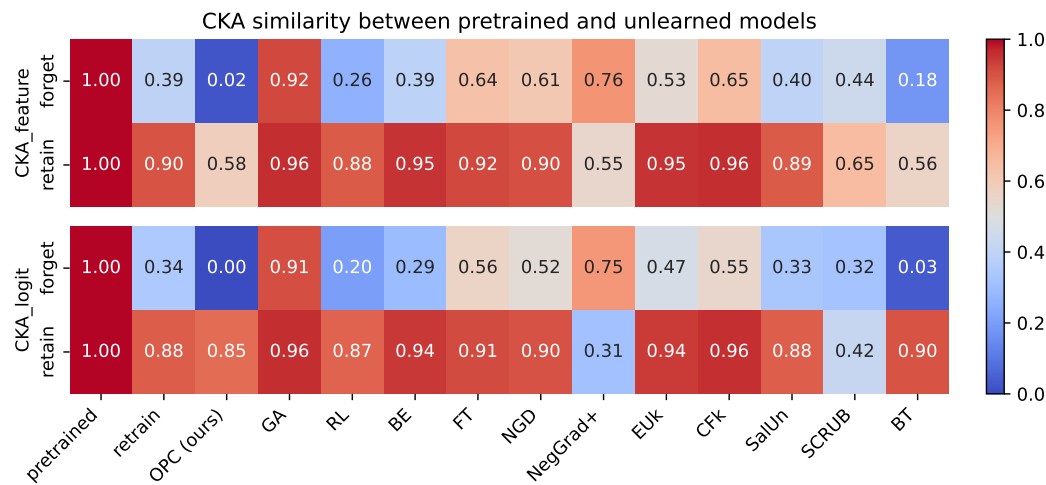

Figure D.2: Visualization of CKA similarity scores between pretrained model and unlearned model, evaluated on SVHN, 30% Class unlearning scenario.

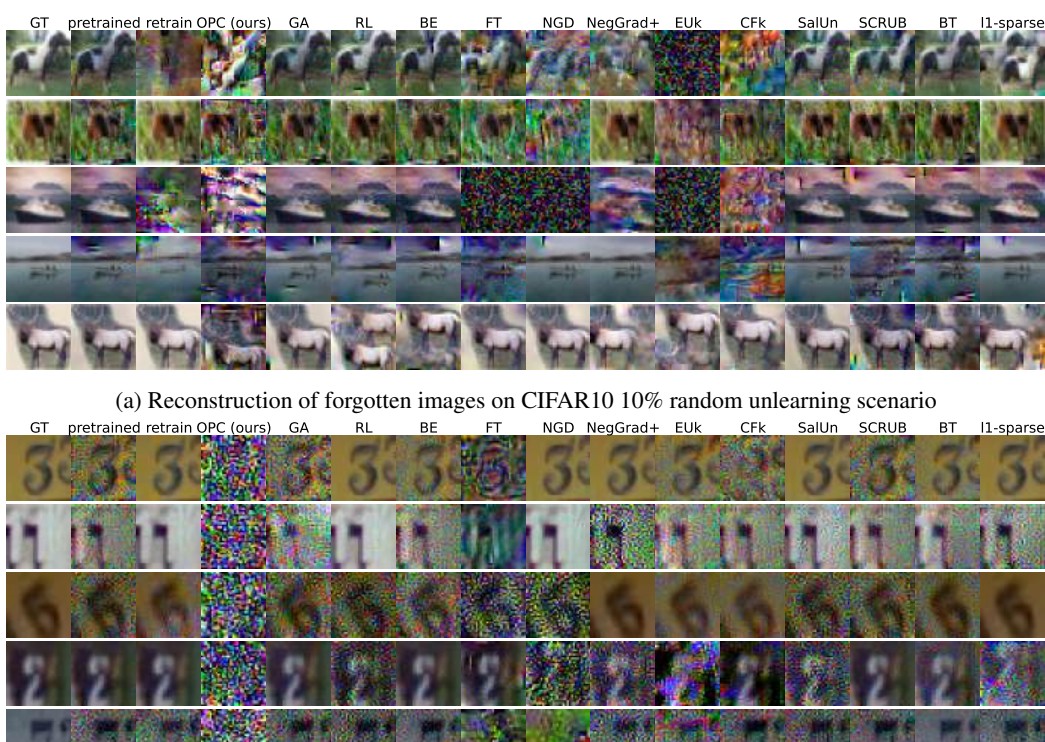

(a) Reconstruction of forgotten images on CIFAR10 10% random unlearning scenario

(b) Reconstruction of forgotten images on SVHN 10% random unlearning scenario

Figure D.3: The results of unlearning inversion. The target images are sampled from the forget set $\mathcal{D}_f$ under 10% random unlearning scenario. GT represents the ground truth image from the dataset and others are the results of inversion attacks from each unlearned model.

The main observation is consistent to the class unlearning scenario, that the forget features of **OPC** is less similar, and the retain features are close to the pretrained model. The CKA similarity score of **OPC** on CIFAR10 is quite larger than other scenarios, but still significantly smaller than the benchmark methods.

Unlike the class unlearning scenario, benchmark unlearning methods extremely high similarity and near-zero gap was observed between the forget feature and retain features.

This may evident the forgetting is failed on almost all methods, while only **OPC** succeeded.

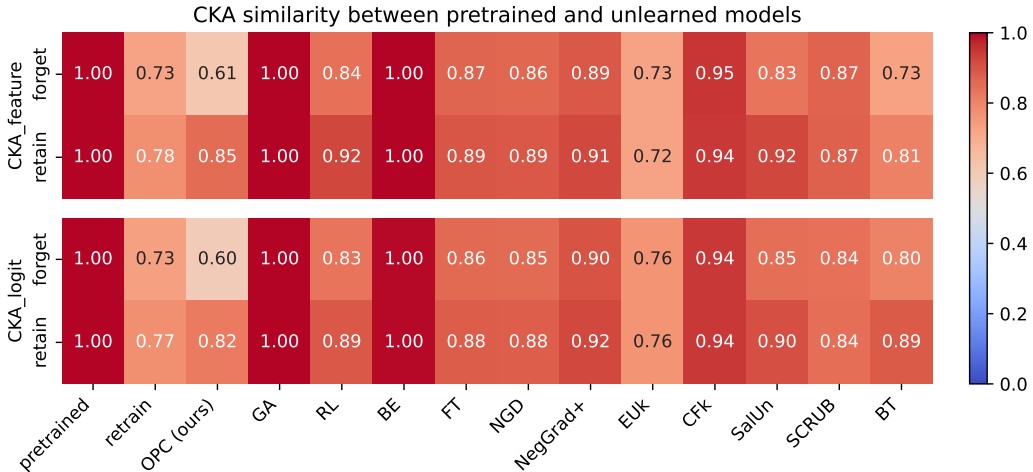

(a) Evaluation result on CIFAR10.

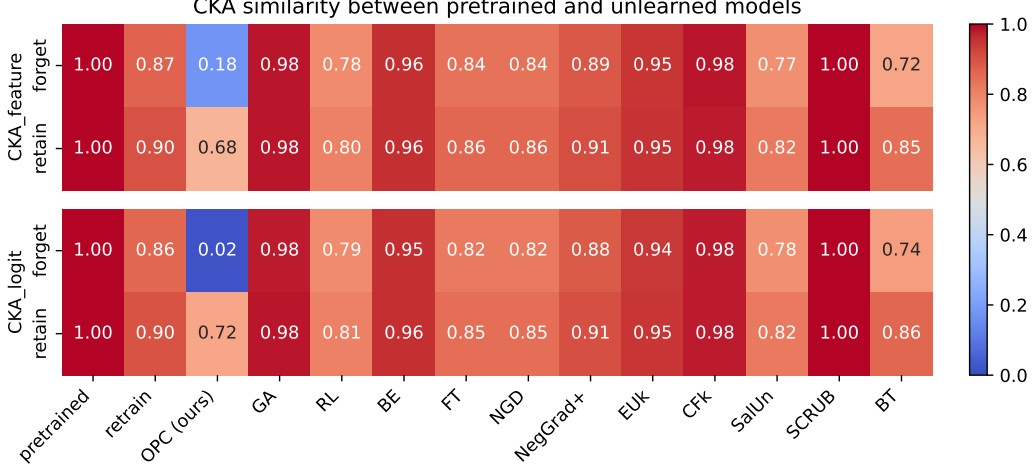

(b) Evaluation result on SVHN.

Figure D.4: Visualization of CKA similarity scores between pretrained model and unlearned model, evaluated on 10% random unlearning scenario. CKA-feature and CKA-logit represent the CKA score computed on $f_\theta(x)$ and $\mathbf{m}_\theta$ respectively.

### D.5.3 RECOVERY ATTACK RESULTS

We applied the least-square based recovery attack, the FM-recovery, on random unlearning scenario. The recovered UA scores are depicted in Fig. D.5 and detailed results of feature mapping recovery are shown in Table D.4. The results of head recovery attack are in Table D.5

Unlike the class unlearning, the significant recovery was not observed on benchmark unlearning methods, due to their severe under-forgetting.

The performance recovery was observed on **OPC**, but we emphasize that the recovered forget accuracy is still advantageous in forgetting, compared to all other unlearning methods.

Table D.4: Recovered performance with $W^*$ and pretrained head on 10% random unlearning scenario

| CIFAR10 | $\mathcal{D}_f$ | $\mathcal{D}_r$ | $\mathcal{D}_{test}$ | MIA$^e$ | SVHN | $\mathcal{D}_f$ | $\mathcal{D}_r$ | $\mathcal{D}_{test}$ | MIA$^e$ |
|---|---|---|---|---|---|---|---|---|---|
| Pretrained | 99.356 | 99.432 | 94.520 | 0.015 | Pretrained | 99.151 | 99.334 | 92.736 | 0.015 |
| Retrain | 90.489 | 99.570 | 89.110 | 0.172 | Retrain | 92.826 | 99.978 | 92.390 | 0.141 |
| OPC (ours) | 87.956 | 99.422 | 91.970 | 0.271 | OPC (ours) | 69.862 | 99.184 | 92.225 | 0.913 |
| GA(Thudi et al., 2022) | 99.311 | 99.430 | 94.340 | 0.018 | GA(Thudi et al., 2022) | 98.878 | 99.316 | 92.498 | 0.016 |
| RL(Golatkar et al., 2020) | 94.000 | 99.916 | 93.960 | 0.194 | RL(Golatkar et al., 2020) | 92.356 | 96.153 | 91.772 | 0.125 |
| BE(Chen et al., 2023) | 99.333 | 99.437 | 94.380 | 0.016 | BE(Chen et al., 2023) | 99.135 | 99.287 | 92.221 | 0.015 |
| FT(Warnecke et al., 2023) | 95.511 | 99.728 | 93.200 | 0.114 | FT(Warnecke et al., 2023) | 93.872 | 99.643 | 94.211 | 0.099 |
| NGD(Chourasia & Shah, 2023) | 96.000 | 99.731 | 93.540 | 0.114 | NGD(Chourasia & Shah, 2023) | 94.373 | 99.589 | 94.353 | 0.092 |
| NegGrad+(Kurmanji et al., 2023) | 96.133 | 99.770 | 93.210 | 0.109 | NegGrad+(Kurmanji et al., 2023) | 94.449 | 99.916 | 93.977 | 0.100 |
| EUk(Goel et al., 2022) | 99.133 | 99.694 | 93.600 | 0.041 | EUk(Goel et al., 2022) | 97.952 | 99.975 | 92.425 | 0.059 |
| CFk(Goel et al., 2022) | 99.311 | 99.842 | 94.080 | 0.028 | CFk(Goel et al., 2022) | 99.151 | 99.993 | 92.836 | 0.022 |
| SalUn(Fan et al., 2024) | 93.889 | 99.896 | 93.810 | 0.200 | SalUn(Fan et al., 2024) | 92.143 | 97.695 | 91.580 | 0.137 |
| SCRUB(Kurmanji et al., 2023) | 99.400 | 99.541 | 94.230 | 0.025 | SCRUB(Kurmanji et al., 2023) | 99.151 | 99.388 | 92.717 | 0.014 |
| BT(Chundawat et al., 2023) | 93.000 | 99.351 | 93.150 | 0.193 | BT(Chundawat et al., 2023) | 96.041 | 99.196 | 91.848 | 0.159 |
| $l1$-sparse(Jia et al., 2023) | 94.089 | 98.309 | 92.020 | 0.110 | $l1$-sparse(Jia et al., 2023) | 93.781 | 98.910 | 93.147 | 0.103 |

Table D.5: Recovered performance with head recovery on 10% random unlearning scenario

| CIFAR10 | $\mathcal{D}_f$ | $\mathcal{D}_r$ | $\mathcal{D}_{test}$ | MIA$^e$ | SVHN | $\mathcal{D}_f$ | $\mathcal{D}_r$ | $\mathcal{D}_{test}$ | MIA$^e$ |
|---|---|---|---|---|---|---|---|---|---|
| Pretrained | 99.644 | 99.575 | 94.400 | 0.094 | Pretrained | 99.287 | 99.441 | 92.663 | 0.149 |
| Retrain | 90.578 | 99.704 | 89.120 | 0.332 | Retrain | 92.765 | 99.998 | 92.033 | 0.271 |
| OPC (ours) | 87.156 | 99.610 | 92.050 | 0.512 | OPC (ours) | 40.983 | 99.933 | 92.371 | 1.000 |
| GA(Thudi et al., 2022) | 99.444 | 99.560 | 94.290 | 0.094 | GA(Thudi et al., 2022) | 98.908 | 99.385 | 92.244 | 0.153 |
| RL(Golatkar et al., 2020) | 93.689 | 99.968 | 93.850 | 0.360 | RL(Golatkar et al., 2020) | 91.506 | 95.713 | 91.000 | 0.405 |
| BE(Chen et al., 2023) | 99.622 | 99.565 | 94.390 | 0.096 | BE(Chen et al., 2023) | 99.257 | 99.405 | 91.887 | 0.169 |
| FT(Warnecke et al., 2023) | 95.711 | 99.812 | 93.060 | 0.227 | FT(Warnecke et al., 2023) | 94.267 | 99.988 | 94.353 | 0.213 |
| NGD(Chourasia & Shah, 2023) | 96.089 | 99.807 | 93.610 | 0.238 | NGD(Chourasia & Shah, 2023) | 94.616 | 99.992 | 94.472 | 0.213 |
| NegGrad+(Kurmanji et al., 2023) | 96.378 | 99.840 | 93.390 | 0.227 | NegGrad+(Kurmanji et al., 2023) | 94.130 | 99.981 | 93.665 | 0.248 |
| EUk(Goel et al., 2022) | 99.178 | 99.867 | 93.630 | 0.152 | EUk(Goel et al., 2022) | 97.877 | 99.990 | 92.179 | 0.196 |
| CFk(Goel et al., 2022) | 99.422 | 99.956 | 94.150 | 0.114 | CFk(Goel et al., 2022) | 99.302 | 99.990 | 92.406 | 0.173 |
| SalUn(Fan et al., 2024) | 93.689 | 99.963 | 93.920 | 0.342 | SalUn(Fan et al., 2024) | 91.066 | 97.481 | 90.731 | 0.429 |
| SCRUB(Kurmanji et al., 2023) | 99.400 | 99.627 | 94.130 | 0.103 | SCRUB(Kurmanji et al., 2023) | 99.257 | 99.508 | 92.628 | 0.148 |
| BT(Chundawat et al., 2023) | 92.089 | 99.435 | 93.180 | 0.377 | BT(Chundawat et al., 2023) | 93.159 | 98.773 | 90.988 | 0.566 |
| $l1$-sparse(Jia et al., 2023) | 93.933 | 98.358 | 91.960 | 0.200 | $l1$-sparse(Jia et al., 2023) | 93.523 | 98.970 | 92.601 | 0.279 |

## D.6 DIFFUSION MODELS

Here, we present the performance of individual targets, complementing the averaged results shown in Table 5. Detailed results can be found in Tables D.6 and D.7. As illustrated in Fig. D.6, **OPC** effectively preserves performance in both style and object for the retain prompts.

## E ADDITIONAL EVALUATIONS

In this section, we present additional experiments conducted to demonstrate the scalability of **OPC** across different models and datasets. For the alternative model architecture, we selected ViT Dosovitskiy et al. (2021), specifically ViT-B-32, to reduce computational overhead. As alternative dataset, we chose TinyImageNet Le & Yang (2015), which contain a larger number of classes and data samples.

Similar to results with ResnNet-18 on CIFAR and SVHN, **OPC** outperforms the benchmark methods and shows resistance on recovery attacks. Unfortunately, the unlearning inversion attack was not feasible since Hu et al. (2024) implementation did not work with ViT.

### E.1 TINYIMAGENET WITH VIT

For the experimental setup, we selected three unlearning algorithms: **FT**, **RL**, and **SalUn**, from those used in Section 4.1, and additionally included Selective Synaptic Dampening (**SSD**), a method that incorporates ViT. **SSD** performs unlearning by dampening weights that have a higher impact on the Fisher information of the forget set compared to the rest of the dataset Foster et al. (2024). For data augmentation, we applied RandomCrop(64, 4) and RandomHorizontalFlip, from the torchvisionmaintainers & contributors (2016) library.

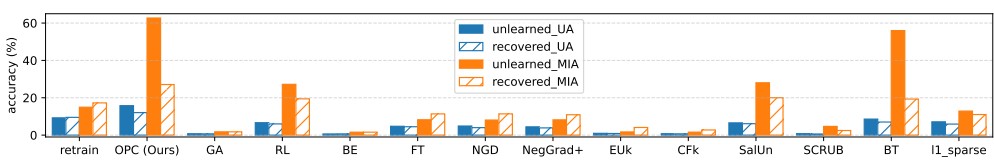

(a) Results of recovery attack on CIFAR10 10% random unlearning scenario

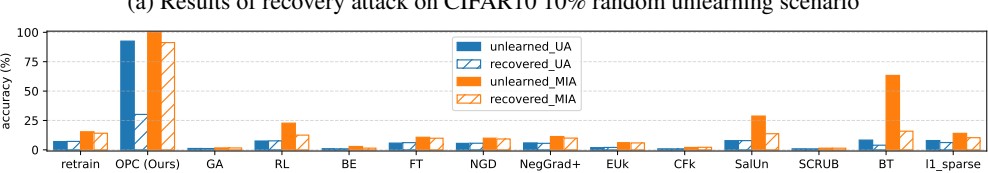

(b) Results of recovery attack on SVHN 10% random unlearning scenario

Figure D.5: UA and MIA score of unlearned model and FM-recovered model.

Table D.6: Individual performance of SD on UnlearnCanvas object unlearning scenario

| Object | UA | IRA | CRA | FID | Object | UA | IRA | CRA | FID |
|---|---|---|---|---|---|---|---|---|---|
| Architectures | 87.843 | 98.617 | 94.922 | 56.8003 | Horses | 99.608 | 98.122 | 95.373 | 56.5258 |
| Bears | 95.294 | 98.555 | 96.863 | 54.3786 | Human | 74.51 | 98.596 | 91.196 | 66.3477 |
| Birds | 98.824 | 98.679 | 92.98 | 61.7023 | Jellyfish | 100 | 98.431 | 96.882 | 55.8011 |
| Butterfly | 99.216 | 98.163 | 94.843 | 58.502 | Rabbits | 100 | 98.39 | 96.02 | 54.0326 |
| Cats | 96.863 | 97.771 | 91.843 | 61.6043 | Sandwiches | 98.039 | 98.246 | 97 | 56.6227 |
| Dogs | 99.216 | 98.246 | 97.255 | 52.869 | Sea | 91.373 | 98.369 | 96.02 | 53.081 |
| Fishes | 94.902 | 98.885 | 95.686 | 53.2249 | Statues | 100 | 98.597 | 97.451 | 52.6221 |
| Flame | 94.51 | 98.122 | 96.118 | 55.6122 | Towers | 99.608 | 98.514 | 96.412 | 54.6292 |
| Flowers | 94.902 | 98.638 | 97.569 | 54.8717 | Trees | 85.49 | 98.184 | 93.863 | 57.8076 |
| Frogs | 100 | 97.957 | 97.569 | 55.0085 | Waterfalls | 99.608 | 98.514 | 96.417 | 54.768 |

Details on training procedures and runtime task are provided in Table E.1. On 10% class unlearning scenario, the additional hyperparameters used were as follows: for **OPC**, $\{coeff\_ce: 1, coeff\_un: 0.05\}$, for **SalUn**, $\{pt: 0.5\}$; and for **SSD**, $\{dampening\_constant: 0.4, size\_scaler: 4.2\}$. On 10% element unlearning scenario, for **OPC**, $\{coeff\_ce: 1, coeff\_un: 0.07\}$, for **SalUn**, $\{pt: 0.5\}$; and for **SSD**, $\{dampening\_constant: 0.1, size\_scaler: 2\}$. The hyperparameters for **SSD** follow the implementation described in Foster et al. (2024). The batch size was limited to 128 due to VRAM constraints. The optimizer used in our experiments was PyTorch's AdamW with a weight decay of 0.3. For learning rate scheduling, we employed PyTorch's CosineAnnealingLR with a $T\_max$ value of the train's epoch, and a $eta\_min$ value of $\frac{1}{100}$ of initial learning rate on pre-training and 0 on unlearning.

Unlike the approach described in Section C.1, the pretrained models used here were fine-tuned from ImageNet-pretrained weights with initial learning rate of $1e-5$ and 5 epochs, following the methodology in Foster et al. (2024). As a result, in the context of unlearning on TinyImageNet, retraining is no longer considered a prohibitively costly method, and cannot be the gold standard of exact unlearning anymore. Consequently, only the efficacy of forgetting is desirable regardless the training cost, compared to the retraining, in TinyImageNet forgetting benchmark.

### E.1.1 CKA SIMILARITY

We first analyze the CKA similarity compared to the pretrained model. As depicted in Fig. E.1, the results are consistent to the ResNet-18 results. The CKA similarities of forget features are still large on benchmark unlearned models, while **OPC**-unleared model shows near-zero similarity. On retrain set $\mathcal{D}_r$, all models including **OPC** shows higher similarity.

The results on random unlearning scenario is similar to CIFAR10 result on random unlearning. but however **OPC** show significantly different forget features compared to the benchmark unlearning methods.

Table D.7: Individual performance of SD on UnlearnCanvas style unlearning scenario

| Style | UA | IRA | CRA | FID | Style | UA | IRA | CRA | FID |
|---|---|---|---|---|---|---|---|---|---|
| Abstractionism | 100 | 94.92 | 98.039 | 56.3459 | Magic Cube | 100 | 97.62 | 97.627 | 54.0653 |
| Artist Sketch | 94 | 97.06 | 98.588 | 53.9738 | Meta Physics | 96 | 97.46 | 98.471 | 53.4481 |
| Blossom Season | 100 | 96.9 | 98.353 | 54.2129 | Meteor Shower | 99 | 96.48 | 98.196 | 52.6702 |
| Bricks | 100 | 95.84 | 98.588 | 54.6203 | Monet | 100 | 96.98 | 98.118 | 53.896 |
| Byzantine | 99 | 98.12 | 98.02 | 53.9902 | Mosaic | 100 | 97 | 98.627 | 52.8538 |
| Cartoon | 95 | 95.92 | 98.275 | 54.3846 | Neon Lines | 97 | 96.94 | 98.196 | 53.8218 |
| Cold Warm | 98 | 98.68 | 98.039 | 55.1618 | On Fire | 98 | 97.62 | 98.392 | 57.3748 |
| Color Fantasy | 100 | 98.02 | 98.333 | 56.4323 | Pastel | 100 | 97.18 | 98.765 | 53.5829 |
| Comic Etch | 100 | 98.58 | 98.529 | 54.8655 | Pencil Drawing | 95 | 97.44 | 98.275 | 55.016 |
| Crayon | 100 | 97.64 | 98.216 | 54.6655 | Picasso | 100 | 97.16 | 98.627 | 52.8177 |
| Cubism | 97 | 94.78 | 98.314 | 59.1373 | Pop Art | 99 | 92.86 | 98.392 | 58.534 |
| Dadaism | 100 | 97.5 | 97.765 | 55.4235 | Red Blue Ink | 100 | 97 | 98.667 | 54.7548 |
| Dapple | 100 | 96.82 | 98.667 | 52.3902 | Rust | 100 | 97.14 | 98.706 | 55.7999 |
| Defoliation | 99 | 97.34 | 98.471 | 53.3461 | Sketch | 99 | 97.68 | 98.137 | 54.6318 |
| Early Autumn | 95 | 97.16 | 98.784 | 53.8521 | Sponge Dabbed | 100 | 97.14 | 98.333 | 55.0828 |
| Expressionism | 100 | 96.62 | 98.353 | 53.5721 | Structuralism | 96 | 97.4 | 98.412 | 55.3737 |
| Fauvism | 100 | 96.64 | 98.098 | 56.275 | Superstring | 100 | 97.92 | 98.275 | 54.4378 |
| French | 100 | 98.04 | 98.137 | 52.7762 | Surrealism | 94 | 93.6 | 96.941 | 54.6372 |
| Glowing Sunset | 96 | 97.9 | 98.784 | 54.3242 | Ukiyoe | 100 | 97.72 | 98.627 | 55.0374 |
| Gorgeous Love | 100 | 97.14 | 98.627 | 53.756 | Van Gogh | 100 | 96.52 | 98.392 | 55.4781 |
| Greenfield | 97 | 97.92 | 98.49 | 53.3042 | Vibrant Flow | 100 | 97.74 | 98.647 | 55.3895 |
| Impressionism | 100 | 98.54 | 98.392 | 54.4472 | Warm Love | 99 | 97.84 | 98.647 | 52.9688 |
| Ink Art | 97 | 97.64 | 98.294 | 54.4843 | Warm Smear | 96 | 96.8 | 98.569 | 55.6418 |
| Joy | 99 | 93.36 | 98.588 | 58.7899 | Watercolor | 82 | 96.92 | 98.412 | 56.0173 |
| Liquid Dreams | 94 | 97.4 | 98.902 | 53.4098 | Winter | 65 | 97.5 | 99 | 53.2705 |

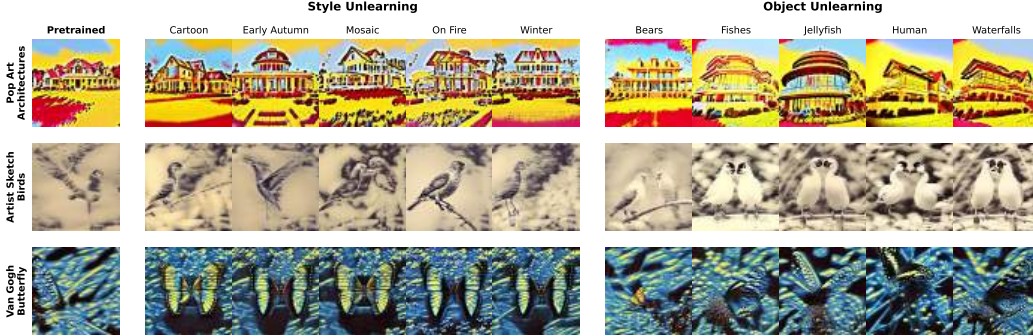

Figure D.6: Image generation from the retain prompts in UnlearnCanvas. The row names correspond to the target prompts, while the column names indicate the unlearn targets.

### E.1.2 RECOVERY ATTACK RESULTS

We applied least square-based recovery attack on ViT with TinyImageNet, and provide the results in **??** and Table E.3, and visualize in Fig. E.2.

In class unlearning scenario, almost all benchmarks show the vulnerability. Similar to ResNet-18 experiments, almost all unlearned models except **OPC**, were recovered its performance under both feature mapping attack and head recovery attack. The retraining shows minor resistance, but the retrained features of forget samples were informative enough to recover the model performance.

Results on random unlearning, does not show the recovery, as forgetting on all unlearning process were imperfect and there's nothing to recover. However, similar to ResNet-18, the recovered performance of **OPC** is still superior to all others that **OPC** forgets more.

### E.1.3 UNLEARNING PERFORMANCE

The unlearning performances summarized in Table E.4. Compared to the benchmark methods, **OPC** show superior results in both class unlearning and random unlearning scenario. Similar to results

Table E.1: Table of training information on TinyImageNet

| Class 10% | Epochs | Learning rate | Element 10% | Epochs | Learning rate |
|---|---|---|---|---|---|
| Retrain | 5 | 0.0001 | Retrain | 5 | 0.00008 |
| OPC (ours) | 5 | 0.0001 | OPC (ours) | 10 | 0.00002 |
| RL(Golatkar et al., 2020) | 10 | 0.00008 | RL(Golatkar et al., 2020) | 5 | 0.00001 |
| FT(Warnecke et al., 2023) | 15 | 0.0001 | FT(Warnecke et al., 2023) | 5 | 0.00004 |
| SSD(Foster et al., 2024) | Train-Free | Train-Free | SSD(Foster et al., 2024) | Train-Free | Train-Free |
| SalUn(Fan et al., 2024) | 10 | 0.00008 | SalUn(Fan et al., 2024) | 5 | 0.000008 |

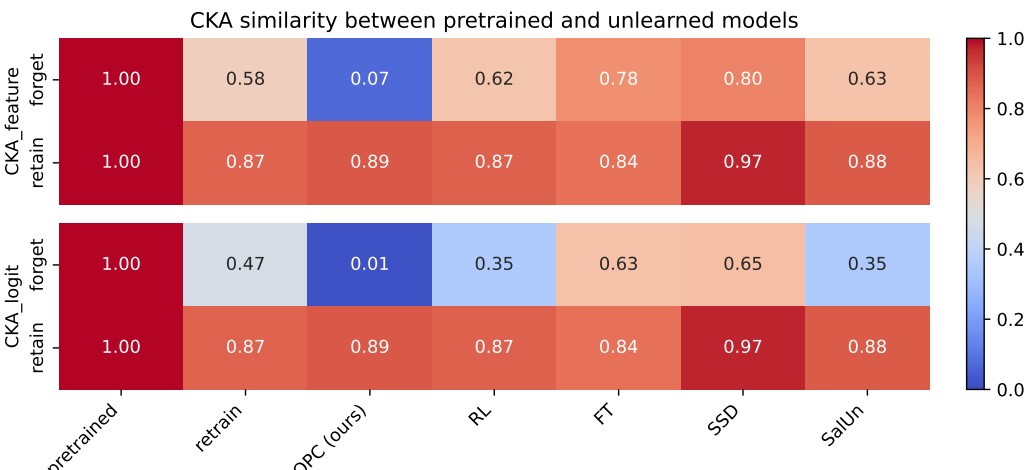

(a) Evaluation result on 10% class unlearning scenario.

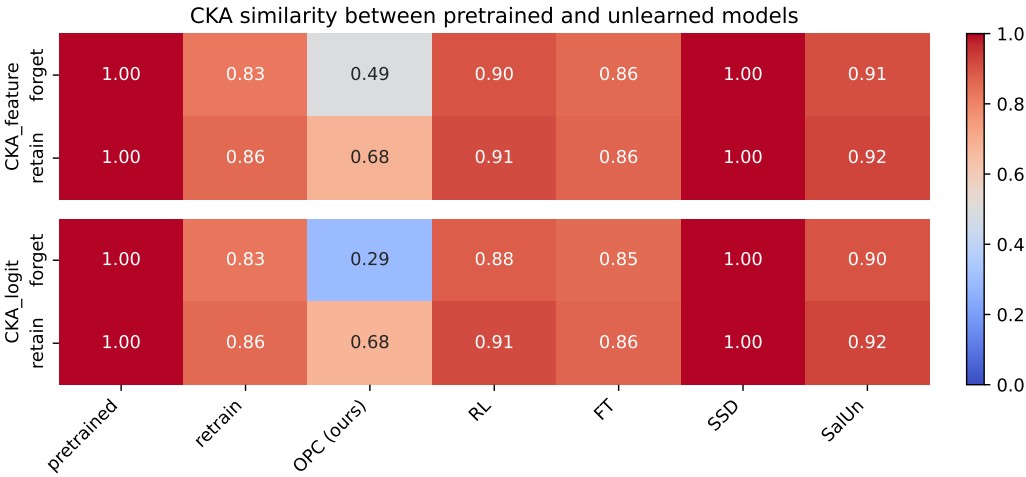

(b) Evaluation result on 10% random unlearning scenario.

Figure E.1: Visualization of CKA similarity scores between pretrained model and unlearned model, evaluated on TinyImageNet. CKA-feature and CKA-logit represent the CKA score computed on $f_\theta(x)$ and $\mathbf{m}_\theta$ respectively.

with ResNet-18, although the forget features are still informative, the performance measurements cannot catch the shallowness forgetting.

Table E.3: Recovered performance with head recovery on TinyImageNet

| Class 10% | Train $\mathcal{D}_f$ | Train $\mathcal{D}_r$ | test $\mathcal{D}_f$ | test $\mathcal{D}_r$ | MIA$^e$ | Element 10% | $\mathcal{D}_f$ | $\mathcal{D}_r$ | $\mathcal{D}_{test}$ | MIA$^e$ |
|---|---|---|---|---|---|---|---|---|---|---|
| Pretrained | 97.230 | 96.139 | 93.600 | 94.288 | 0.283 | Pretrained | 96.230 | 96.296 | 84.237 | 0.303 |
| Retrain | 70.720 | 94.082 | 92.000 | 93.888 | 0.756 | Retrain | 85.890 | 97.749 | 85.497 | 0.354 |
| OPC (ours) | 31.820 | 98.459 | 36.800 | 93.265 | 1.000 | OPC (ours) | 81.370 | 99.407 | 81.236 | 0.863 |
| RL(Golatkar et al., 2020) | 91.760 | 99.626 | 90.600 | 93.532 | 0.992 | RL(Golatkar et al., 2020) | 93.250 | 97.533 | 83.497 | 0.542 |
| FT(Warnecke et al., 2023) | 80.040 | 99.688 | 88.800 | 92.265 | 0.564 | FT(Warnecke et al., 2023) | 88.930 | 99.576 | 81.076 | 0.335 |
| SSD(Foster et al., 2024) | 83.870 | 95.408 | 92.200 | 94.021 | 0.776 | SSD(Foster et al., 2024) | 96.180 | 96.211 | 83.957 | 0.286 |
| SalUn(Fan et al., 2024) | 91.330 | 99.587 | 90.600 | 93.643 | 0.984 | SalUn(Fan et al., 2024) | 94.270 | 97.448 | 83.497 | 0.492 |

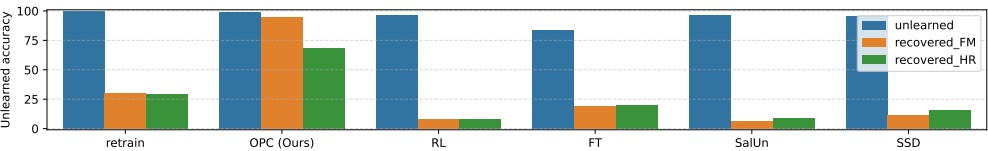

(a) Results of recovery attack on 10% class unlearning scenario

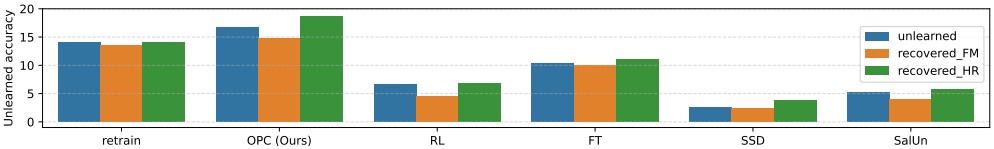

(b) Results of recovery attack on 10% random unlearning scenario

Figure E.2: Recovered UA scores (higher means the unlearning method is more resistant to recovery attack) on TinyImageNet with feature map alignment (FM, orange) and head recovery (HR, green), compared to unlearned UA (which should be 100 for a well-performing unlearning method).

Table E.4: Unlearning performance on TinyImageNet

| Class 10% | Train $\mathcal{D}_f$ | Train $\mathcal{D}_r$ | test $\mathcal{D}_f$ | test $\mathcal{D}_r$ | MIA$^e$ | Element 10% | $\mathcal{D}_f$ | $\mathcal{D}_r$ | $\mathcal{D}_{test}$ | MIA$^e$ | MIA$^p$ |
|---|---|---|---|---|---|---|---|---|---|---|---|
| Pretrained | 97.830 | 97.541 | 85.200 | 83.685 | 0.105 | Pretrained | 97.520 | 97.576 | 83.837 | 0.119 | 0.604 |
| Retrain | 0.000 | 95.844 | 0.000 | 82.818 | 1.000 | Retrain | 85.930 | 98.682 | 85.337 | 0.276 | 0.606 |
| OPC (ours) | 0.660 | 99.427 | 0.400 | 81.129 | 1.000 | OPC (ours) | 83.330 | 99.776 | 81.276 | 0.724 | 0.654 |
| RL(Golatkar et al., 2020) | 3.690 | 99.953 | 2.200 | 81.974 | 1.000 | RL(Golatkar et al., 2020) | 93.330 | 98.803 | 82.376 | 0.422 | 0.631 |
| FT(Warnecke et al., 2023) | 16.490 | 99.977 | 14.600 | 80.596 | 1.000 | FT(Warnecke et al., 2023) | 89.590 | 99.944 | 80.836 | 0.240 | 0.663 |
| SSD(Foster et al., 2024) | 4.730 | 95.800 | 4.800 | 82.263 | 1.000 | SSD(Foster et al., 2024) | 97.350 | 97.356 | 83.597 | 0.128 | 0.600 |
| SalUn(Fan et al., 2024) | 3.240 | 99.941 | 2.000 | 82.040 | 1.000 | SalUn(Fan et al., 2024) | 94.840 | 98.567 | 82.416 | 0.461 | 0.628 |

Table E.5: Unlearning performance with train-free unlearning on prediction head only

| TinyImageNet | Train $\mathcal{D}_f$ | Train $\mathcal{D}_r$ | test $\mathcal{D}_f$ | Test $\mathcal{D}_r$ | MIA$^e$ |
|---|---|---|---|---|---|
| Pretrained | 97.830 | 97.541 | 85.200 | 83.685 | 0.105 |
| Retrain | 0.000 | 95.844 | 0.000 | 82.818 | 1.000 |
| OPC-TF | 0 | 97.02 | 0 | 84.574 | 1.000 |
| RL-TF | 0 | 96.978 | 0 | 84.197 | 1.000 |

### E.1.4 TRAIN-FREE UNLEARNING

In class unlearning scenario, we could consider the unlearning process without training, by modifying theprediction head only. Table E.5 shows the result that the head-only forgetting without training can achieve near-perfect unlearning scores such as forget accuracy and **MIA**$^e$.

## F FEATURE VISUALIZATION AND ANALYSIS

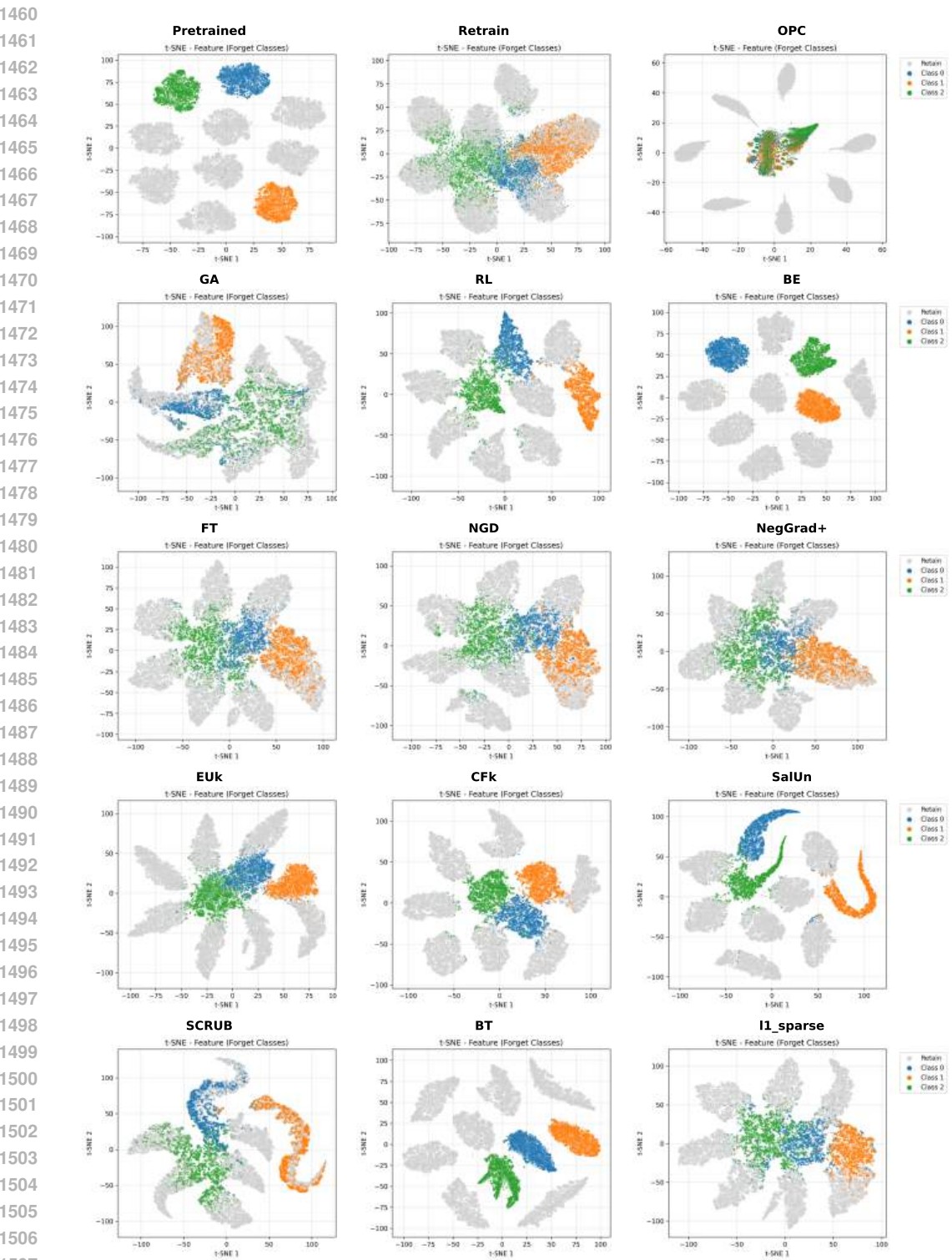

Figure F.1: t-SNE plot of feature on CIFAR10 30% class unlearning scenario

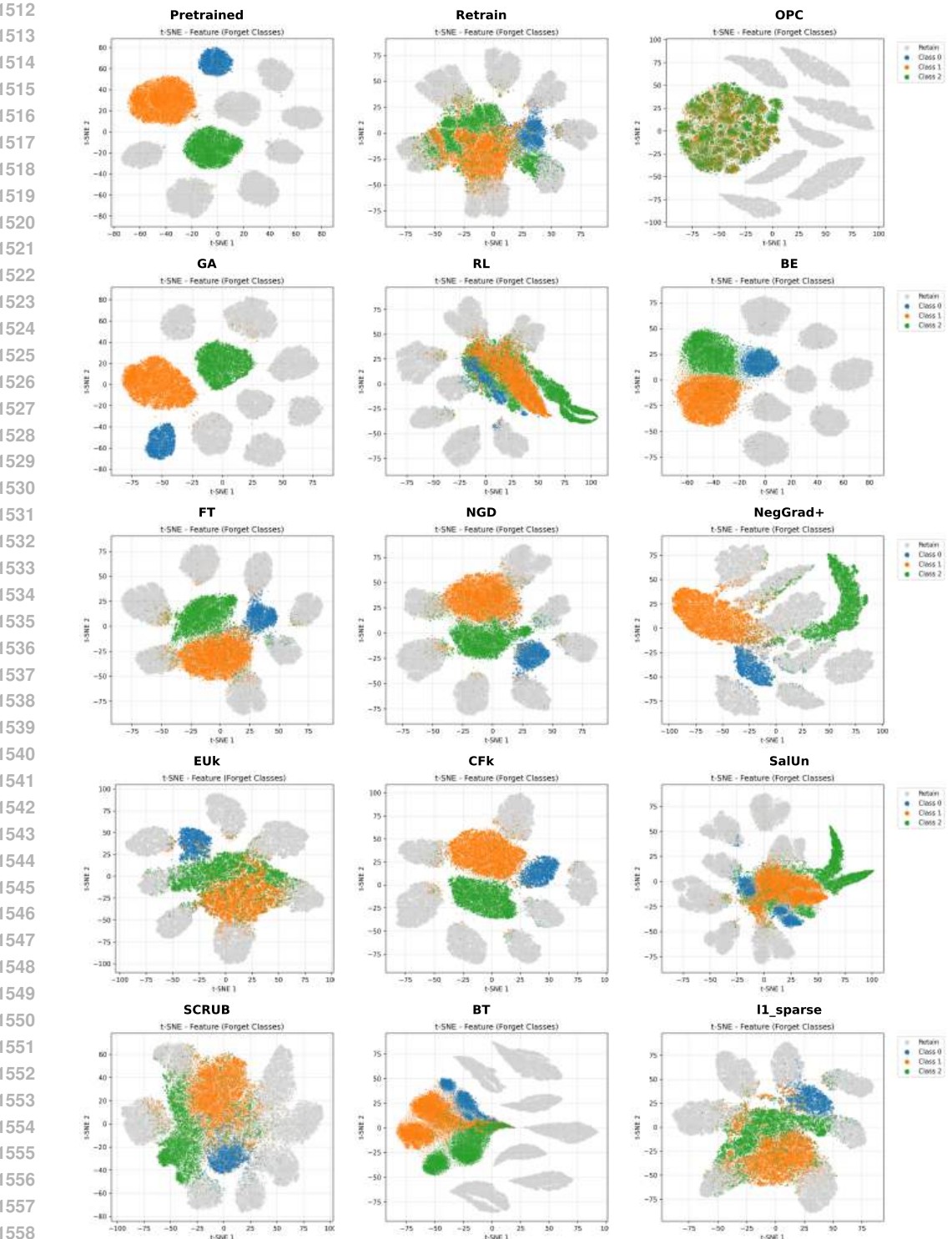

Figure F.2: t-SNE plot of feature on SVHN 30% class unlearning scenario

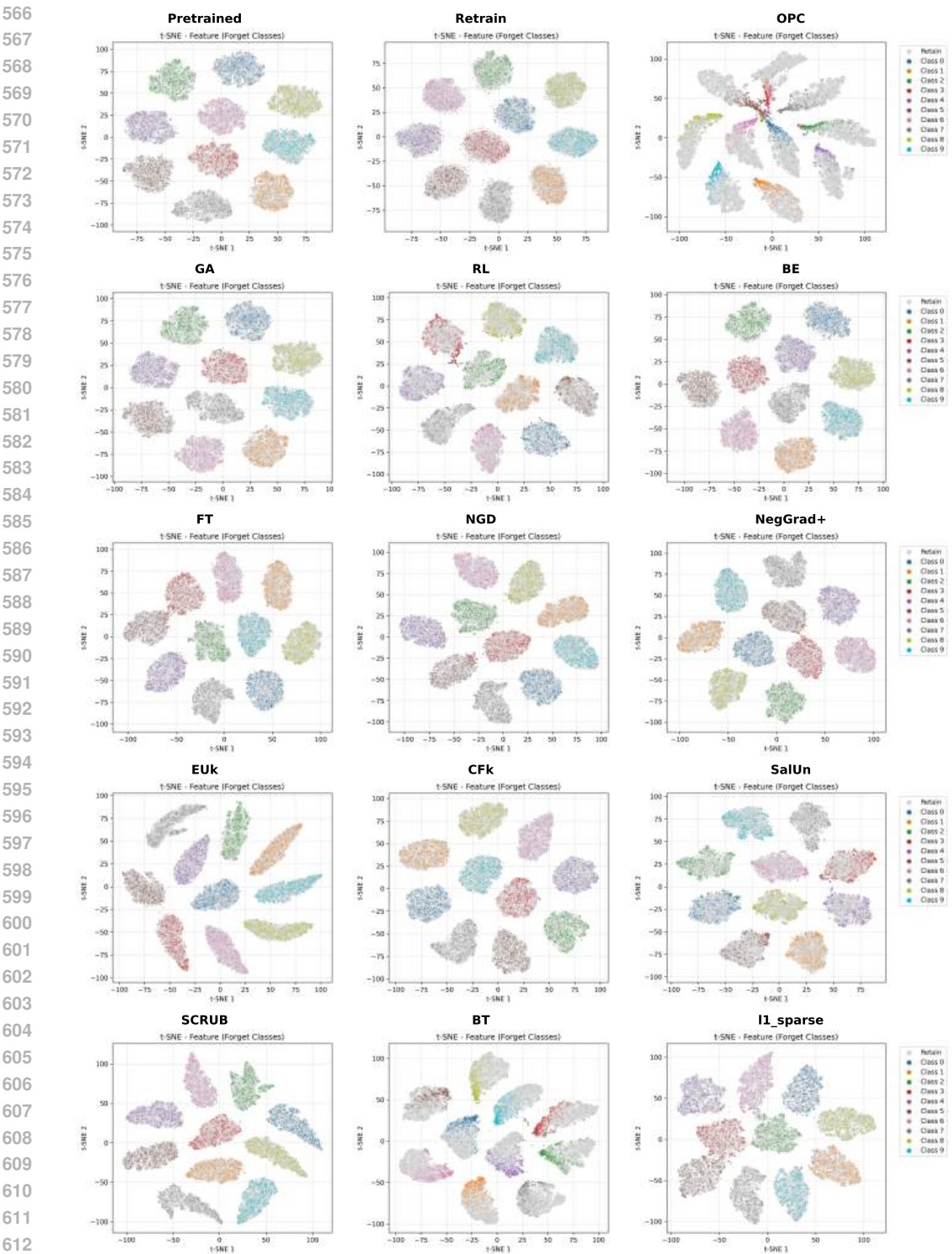

Figure F.3: t-SNE plot of feature on CIFAR10 10% random unlearning scenario

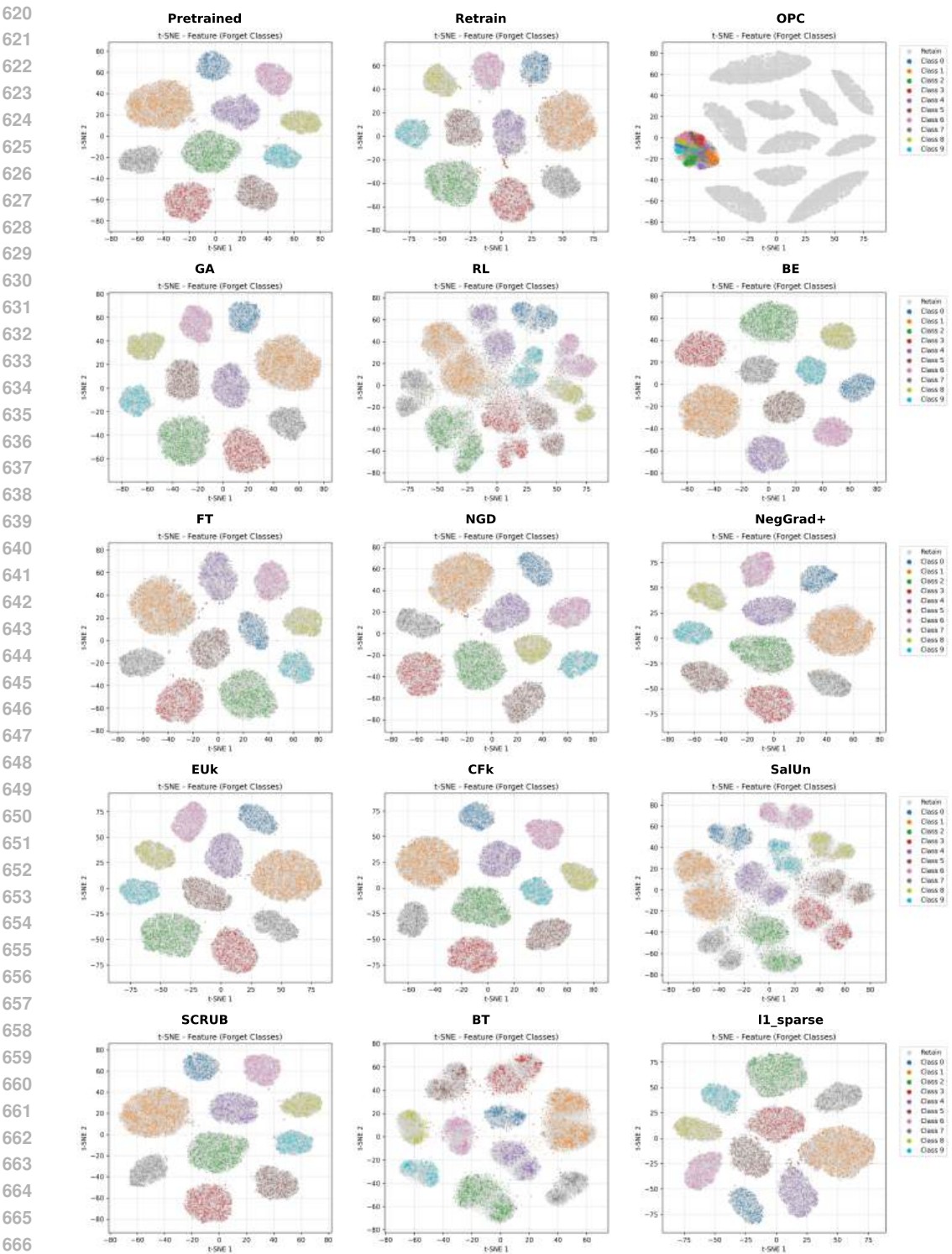

Figure F.4: t-SNE plot of feature on SVHN 10% random unlearning scenario

In Section 4 we investigated the shallowness of forgetting which led by existing MU methods. Interestingly, it was possible to find linear mapping between unlearned feature and pretrained feature, allowing the performance recovery and forget data reconstruction. This recovery indicates that, even after the unlearning was performed, there is a linear separability between the representation clusters of each classes.

Thanks to reviewer, we acknowledge the feature visualization would further provide the intuition on separability on unlearned features and evidence of deep forgetting on information led by OPC unlearning. In this section, we provide tSNE visualization of unlearned features on classifier experiments with explanation.

On class unlearning scenario, in Figs. F.1 and F.2 we observed the separability between forget classes, in almost all unlearned model except OPC. This clear separation of forget features induces the vulnerability of reconstruction and often allow to revert the unlearning process, as shown in Section 4.3. Unlike others, OPC makes unlearned features indistinguishable and hence induce the destruction of class information Note that although the one-point contraction was performed on logits, the class information was removed on features.

On random unlearning scenario, in Figs. F.3 and F.4, all methods except BT and OPC fails to separate the forget samples from representation cluster of correct label. Inherently, the UA and unlearning efficacy are strictly limited, indicating the failure of forgetting.

However, OPC and BT shows different behavior. In CIFAR10 experiment, both OPC and BT makes the inter-class separability between forget features and retain features, inducing larger gain in $MIA^e$ in Table 2. On SVHN results in Fig. F.4, OPC behavior is similar to perfect forgetting on class unlearning scenario, that the forget features are making single cluster and the features are mixed enough to destroy the class information, explaining the remarkable UA gain (92.5% (OPC) vs ¡10% (others) ) while preserving the TA and RA.

# G DISCUSSION ON PREDICTION NORM

## G.1 RELATION BETWEEN FEATURE NORM AND UNCERTAINTY

One motivation of OPC is came from the empirical observation reported in OOD detection literature: the OOD predictions are observed to have smaller norms and larger uncertainty.

In motivation, we found that the retrained model, which is often considered to be the golden standard, shows the consistent behavior of small feature norm and larger entropy on forget dataset. We visualize the entropy of prediction feature norm $\|f_\theta(x)\|_2$ on forget data and retain data in Fig. G.1 with class unlearning scenario on CIFAR10 and SVHN.

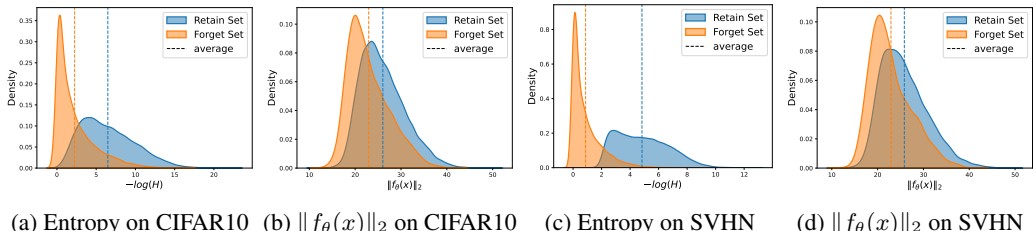

(a) Entropy on CIFAR10   (b) $\|f_\theta(x)\|_2$ on CIFAR10   (c) Entropy on SVHN   (d) $\|f_\theta(x)\|_2$ on SVHN

Figure G.1: The difference of entropy and feature norm of retrained model, on forget dataset and retain dataset. Fig. G.1a and Fig. G.1b are the results from CIFAR10, and Fig. G.1c and Fig. G.1d are the results from SVHN. The forget dataset is consist of 3 classes of each dataset.

We made the larger gap between the forget samples and retain samples on feature norm and entropy, by OPC unlearning algorithm to make forgetting deeper.

## G.2 DISCUSSION ON RELATION BETWEEN BT AND OPC

The BT(Chundawat et al., 2023) shares some behavioral similarities to OPC on logit level, on CKA analysis for class unlearning scenario, and feature visualization in Fig. F.3 on random unlearning scenario with higher $MIA^e$ scores. It sometimes show resistance against the reconstruction attack.

We carefully hypothesize this partial similarity and success of BT is due to small-normed prediction, which induces high uncertainty by Theorem 3.1 and concept of OPC.

Recall that BT employs a knowledge distillation from randomly initialized model, the bad teacher, to guide the broken prediction on forget dataset. Interestingly, the prediction norm from the teacher is consistently low. compared to the pretrained model, as shown in Fig. G.2 with full train dataset. The randomly initialized model gives prediction with much smaller (about 0.01 scale) norm.

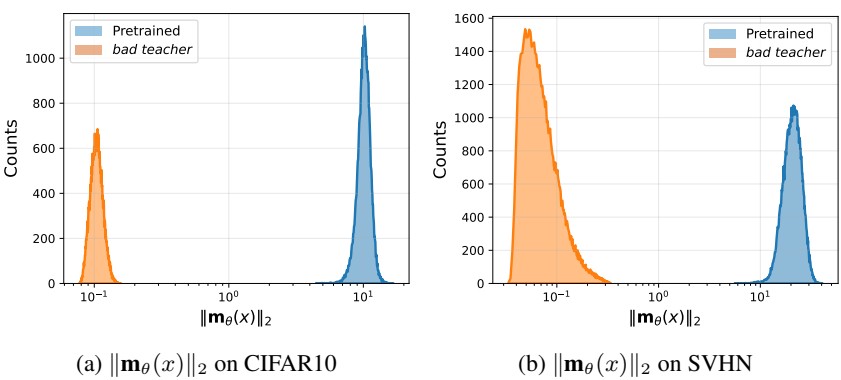

(a) $\|\mathbf{m}_\theta(x)\|_2$ on CIFAR10   (b) $\|\mathbf{m}_\theta(x)\|_2$ on SVHN

Figure G.2: Norm distribution of prediction of *bad teacher* and pretrained model

Since BT-unlearned model is trained to imitate the teacher's behavior, the BT-unlearned model's prediction has small norm too, as depicted in Fig. G.3. Note that the prediction norm was preserved on retain set for both BT and OPC.

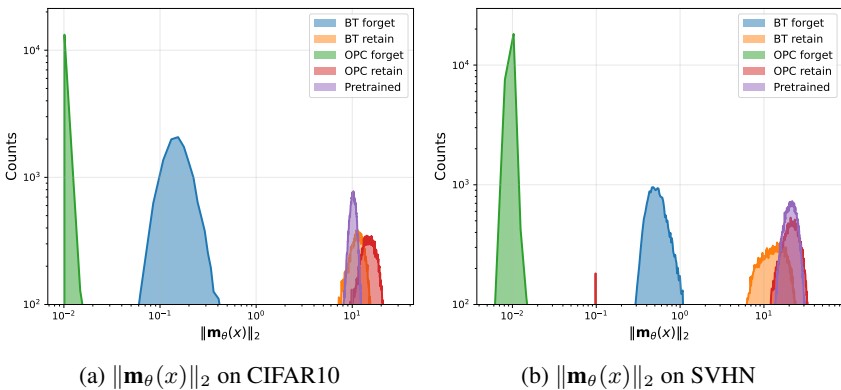

(a) $\|\mathbf{m}_\theta(x)\|_2$ on CIFAR10      (b) $\|\mathbf{m}_\theta(x)\|_2$ on SVHN

Figure G.3: Norm distribution of prediction of $\mathcal{D}_f$ and $\mathcal{D}_r$

However, the magnitude degradation by BT is limited; there is a significant gap between OPC and BT on forget prediction norm, which exhibits the information destruction. Also, the BT training was insufficient to guide the feature-level forgetting, makes the unlearned model vulnerable under recovery or reconstruction attack.

# H    SENSITIVITY ANALYSIS

In this section, we examine the sensitivity of OPC. In H.1, we analyze its sensitivity to hyperparameters. In H.2, we evaluate its scalability across different unlearning scenarios. In H.3, we investigate how the number of iterations affects OPC when the hyperparameters are fixed.

## H.1    HYPERPARAMTER ANALYSIS

Table H.1: Hyperparameter analysis on CIFAR10, 30% class unlearning scenario

| $\mathcal{D}_f$ | $\mathcal{D}_r$ | test $\mathcal{D}_f$ | test $\mathcal{D}_r$ | $\mathbf{MIA}^e$ | $coeff\_ce$ | $coeff\_un$ |
|---|---|---|---|---|---|---|
| 2.037 | 99.737 | 2.333 | 94 | 1 | 1 | 1 |
| 0.037 | 99.771 | 0.167 | 94.371 | 1 | 1 | 0.9 |
| 1.592 | 99.603 | 1.867 | 93.757 | 1 | 1 | 0.8 |
| 0 | 99.606 | 0 | 93.143 | 1 | 1 | 0.7 |
| 3.4 | 99.759 | 3.333 | 93.543 | 1 | 1 | 0.6 |
| 0.044 | 99.791 | 0 | 94.571 | 1 | 1 | 0.5 |
| 1.793 | 99.851 | 1.733 | 94.457 | 1 | 1 | 0.4 |
| 0.852 | 99.908 | 1.033 | 94.814 | 1 | 1 | 0.3 |
| 1.696 | 99.810 | 1.5 | 94.7 | 1 | 1 | 0.2 |
| 25.178 | 99.876 | 23.633 | 94.614 | 1 | 1 | 0.1 |

Table H.1 and Table H.2 present the sensitivity of OPC to the hyperparameters that control the relative contribution of the retain loss and the unlearning loss. Across both the class-unlearning and random-unlearning settings, OPC remains highly stable to the choice of coefficients. In the 30% class-unlearning scenario, decreasing the unlearning coefficient shows that OPC maintains stable retain accuracy and effective forgetting across a broad range of values. However, when the coefficient becomes very small (e.g., 0.1), we observe that the training finishes before the contraction fully occurs, resulting in slightly higher residual forget accuracy. This suggests that extremely small coefficients may under-drive the contraction process.

A similar trend appears in the 10% random-unlearning setting. As the unlearning coefficient $coeff\_un$ increases, the forget accuracy $\mathcal{D}_f$ decreases more aggressively, indicating stronger forgetting. However, a mild trade-off emerges: both $\mathcal{D}_r$ and $\mathcal{D}_{test}$ also drop slightly as the forgetting

Table H.2: Hyperparameter analysis on CIFAR10, 10% random unlearning scenario

| $\mathcal{D}_f$ | $\mathcal{D}_r$ | $\mathcal{D}_{test}$ | $\mathbf{MIA}^e$ | $\mathbf{MIA}^p$ | $coeff\_ce$ | $coeff\_un$ |
|---|---|---|---|---|---|---|
| 65.511 | 98.449 | 89.76 | 0.885 | 0.572 | 0.95 | 0.13 |
| 70.889 | 98.689 | 90.35 | 0.890 | 0.577 | 0.95 | 0.12 |
| 73.556 | 98.859 | 90.5 | 0.883 | 0.579 | 0.95 | 0.11 |
| 74.889 | 99.242 | 91.2 | 0.867 | 0.578 | 0.95 | 0.1 |
| 79.333 | 98.958 | 90.71 | 0.841 | 0.575 | 0.95 | 0.09 |
| 82.156 | 99.316 | 91.4 | 0.819 | 0.576 | 0.95 | 0.08 |
| 85.156 | 99.519 | 92.2 | 0.769 | 0.573 | 0.95 | 0.07 |
| 87.378 | 99.746 | 92.36 | 0.752 | 0.579 | 0.95 | 0.06 |
| 84.244 | 99.190 | 90.930 | 0.627 | 0.570 | 0.95 | 0.05 |
| 90.733 | 99.849 | 92.98 | 0.586 | 0.574 | 0.95 | 0.04 |
| 94.4 | 99.928 | 93.56 | 0.374 | 0.566 | 0.95 | 0.03 |
| 97.089 | 99.985 | 94.07 | 0.251 | 0.567 | 0.95 | 0.02 |
| 98.756 | 99.978 | 94.51 | 0.095 | 0.557 | 0.95 | 0.01 |

strength increases. While this sensitivity is relatively small compared to the overall stability of OPC, it highlights that choosing $coeff\_un$ requires balancing forgetting strength with utility preservation.

## H.2 SCALING ANALYSIS

Table H.3: Scaling analysis on CIFAR10

| Class unlearning scenario | | | | | | | |
|---|---|---|---|---|---|---|---|
| Unit(10%) | $\mathcal{D}_f$ | $\mathcal{D}_r$ | test $\mathcal{D}_f$ | test $\mathcal{D}_r$ | $\mathbf{MIA}^e$ | $coeff\_ce$ | $coeff\_un$ |
| 5 | 0 | 99.702 | 0 | 96.08 | 1 | 1 | 1 |
| 4 | 0.006 | 99.426 | 0 | 94.417 | 1 | 1 | 0.9 |
| 3 | 0 | 99.746 | 0 | 94.129 | 1 | 1 | 0.9 |
| 2 | 0.089 | 99.606 | 0.1 | 93.4125 | 1 | 1 | 0.9 |
| 1 | 0.022 | 99.412 | 0 | 93.167 | 1 | 1 | 0.9 |

| Random unlearning scenario | | | | | | | |
|---|---|---|---|---|---|---|---|
| Unit(10%) | $\mathcal{D}_f$ | $\mathcal{D}_r$ | $\mathcal{D}_{test}$ | $\mathbf{MIA}^e$ | $\mathbf{MIA}^p$ | $coeff\_ce$ | $coeff\_un$ |
| 5 | 89.431 | 99.658 | 89.57 | 0.579 | 0.634 | 1 | 0.15 |
| 4 | 88.578 | 99.7 | 89.98 | 0.602 | 0.625 | 1 | 0.15 |
| 3 | 86.452 | 99.454 | 90.3 | 0.665 | 0.612 | 1 | 0.15 |
| 2 | 79.089 | 98.764 | 89.13 | 0.767 | 0.599 | 1 | 0.15 |
| 1 | 84.244 | 99.190 | 90.930 | 0.627 | 0.570 | 0.95 | 0.05 |

In Table H.3, we present the results of applying OPC across various unlearning scenarios. In both the Class and Random unlearning settings, OPC maintains stable unlearning performance even as the size of the forget set increases, and this is achieved with simple hyperparameter adjustments.

Except for the hyperparameters explicitly shown, all other hyperparameters (with the exception of the number of epochs) follow the configuration in Section C.1. For the Class unlearning scenario, we use 25 epochs only when the forget ratio is 50%, and 30 epochs for all other cases. For the Random unlearning scenario, we use 20 epochs for all experiments.

## H.3 ITERATION ANALYSIS

Similarly, OPC demonstrates robustness to the number of iterations, as shown in Table H.4 and Table H.5. Other unlearning methods often show degraded performance on both $\mathcal{D}_r$ and $\mathcal{D}_f$ as the number of epochs increases, or they even show partial recovery on $\mathcal{D}_f$. In contrast, OPC continues to

Table H.4: Iteration analysis on CIFAR10 30% class unlearning scenario

| Epochs | $\mathcal{D}_f$ | $\mathcal{D}_r$ | test $\mathcal{D}_f$ | test $\mathcal{D}_r$ | $\mathbf{MIA}^e$ | $coeff\_ce$ | $coeff\_un$ |
|--------|-------|--------|--------|--------|------|------|------|
| 40 | 0.067 | 99.838 | 0.1 | 94.229 | 1 | 1 | 0.7 |
| 30 | 0 | 99.606 | 0 | 93.143 | 1 | 1 | 0.7 |
| 20 | 0.215 | 99.635 | 0.167 | 94.114 | 1 | 1 | 0.7 |
| 10 | 2.274 | 99.467 | 1.667 | 93.543 | 1 | 1 | 0.7 |
| 5 | 5.652 | 98.851 | 5.133 | 93.086 | 1 | 1 | 0.7 |
| 1 | 4.548 | 99.248 | 4.4 | 94.386 | 1 | 1 | 0.7 |

Table H.5: Iteration analysis on CIFAR10 10% random unlearning scenario

| Epochs | $\mathcal{D}_f$ | $\mathcal{D}_r$ | $\mathcal{D}_{test}$ | $\mathbf{MIA}^e$ | $\mathbf{MIA}^p$ | $coeff\_ce$ | $coeff\_un$ |
|--------|-------|--------|--------|------|------|------|------|
| 40 | 61.711 | 99.474 | 91.42 | 0.901 | 0.579 | 0.95 | 0.05 |
| 35 | 68.511 | 99.649 | 91.83 | 0.883 | 0.585 | 0.95 | 0.05 |
| 30 | 79 | 99.556 | 91.72 | 0.805 | 0.576 | 0.95 | 0.05 |
| 25 | 85.533 | 99.637 | 92 | 0.764 | 0.581 | 0.95 | 0.05 |
| 20 | 84.244 | 99.190 | 90.930 | 0.627 | 0.570 | 0.95 | 0.05 |
| 15 | 92.844 | 99.864 | 93.19 | 0.513 | 0.562 | 0.95 | 0.05 |
| 10 | 96.022 | 99.906 | 93.62 | 0.258 | 0.561 | 0.95 | 0.05 |
| 5 | 98.6 | 99.862 | 94.33 | 0.131 | 0.559 | 0.95 | 0.05 |
| 1 | 99.178 | 99.531 | 93.84 | 0.062 | 0.551 | 0.95 | 0.05 |

reduce performance on $\mathcal{D}_f$ proportionally to the number of iterations while maintaining performance on $\mathcal{D}_r$.

Due to this behavior, it may appear in Table C.1 that OPC requires a relatively large training budget. However, OPC's training dynamics are highly stable, which allows effective unlearning without any loss in overall performance.

In the random unlearning scenario, the proportional relationship between unlearning performance and the number of iterations is still preserved. However, as shown in Table H.2, $\mathcal{D}_{test}$ gradually decreases, so hyperparameter selection requires careful attention, just as in the case of choosing $coeff_u n$.

