# OpenReview forum: "OPC: One-Point-Contraction Unlearning Toward Deep Feature Forgetting"
_ICLR.cc/2026/Conference — Submitted to ICLR 2026_

### Official Review · Reviewer_g34P · 2025-10-27

**Soundness:** 3
**Presentation:** 3
**Contribution:** 3
**Rating:** 6
**Confidence:** 3

**Summary:**

This paper first identifies the issue of shallow unlearning in most existing methods by conducting a recovery attack. It then proposes One-Point Contraction (OPC), which achieves deep unlearning by normalizing the representation of the forgotten data toward the origin point of the feature space.

**Strengths:**

1. The paper is clearly structured and well-presented.

2. Demonstrates the connection between uncertainty and classification entropy using Theorem 3.1, which clearly motivates the use of the proposed normalization term for unlearning.

3. The extended empirical results, including classification and generative diffusion tasks, consistently show that the proposed method outperforms existing methods in terms of data privacy concerns.

**Weaknesses:**

1. Efficiency and Ablation on OPC Updates: While OPC achieves strong forgetting performance, its efficiency in class unlearning settings is generally lower than other baselines, notably BT, which attains similar results with fewer training iterations.  The paper attributes this to differences in training epochs, yet no ablation or scaling analysis is provided. It would strengthen the work to include systematic ablations showing how OPC’s efficiency and convergence vary with the sizes of $D_f$ and $D_r$, as well as the number of OPC update steps. Such results would clarify whether the slower convergence stems from the contraction mechanism itself or merely from hyperparameter selection.

2. The authors depart from the common unlearning evaluation protocol that compares results against a retrained model as a gold standard. This omission is understandable given OPC’s distinct behavior—collapsing $D_f$ features to the origin rather than erasing them through retraining—but it raises potential generalization and safety concerns. Since OPC enforces degraded responses on $D_f$, the model’s behavior might deviate from retraining even on semantically related test inputs, potentially violating the expected equivalence criterion in unlearning (i.e., consistent predictions with retraining on$D_r$). Discussing these implications would enhance the reader’s understanding of the trade-off between strong forgetting and functional consistency.

3. One stated motivation is to treat $D_f$ as OOD data. However, the paper does not clearly define the degree or notion of OOD induced by OPC. For example, in CIFAR-10 unlearning, should $D_f$ representations behave like pure noise, or like samples from the distribution of SVHN? Making $D_f$ too distinct from the overall data manifold might create new security risks, such as detectable distributional signatures or abnormal uncertainty patterns that could reveal which classes were unlearned. A more explicit discussion of this trade-off between effective forgetting and distributional detectability would be valuable.

**Questions:**

Please refer to the weaknesses.

---

> ### Author Response · Authors · 2025-12-01
>
> We appriciate the reviewer for the constructive feedback and providing the discussion points.
> Below, we address your concerns, with our opinions.
>
> (W1)
>
> First of all, the larger training cost for OPC in class unlearning scenario is due to stability issue on MU methods. We set same training budget to all baselines, approximately 1/3 of retraining epochs. During the training, we experienced the model collapse on retain set and test set, that the TA (test accuracy) and RA (retain accuracy) drops severely. Hence, we saved all checkpoint, chose the best one, and report the epoch number of the best checkpoint. Therefore the most unstable method is showing the best efficiency. On class unlearning, OPC did not show any collapse but convergence, thus no early stopping was required and best results were at the final stage of the training.
>
> However, we would add the sensitivity analysis to the revised manuscript. The results are following expectations: stronger minimization of OPC loss yield more forgetting, damage on test set but with smoother degradation (e.g. 35% UA with 1% TA drop on CIFAR10 random 10% forgetting), and faster convergence.
>
> (W2)
>
> We thank the reviewer for the the considerable point. As you pointed, we don't aim to imitate the retrained model, but focus on original requirement: unlearned model should work well on retain set and test set, while it collapses selectively on forget set.
>
> If we apply OPC unlearning moderately, OPC also can mimic the retrained model on accuracies, MIA, feature norm and entropy distribution. However, we regard this goal may not meet the practical requirement. If the unlearning shows 8% UA due to randomly distributed forget query and developer say ">90% of your data are not forgotten, due to generalization", the data provider would not be satisfied.
>
> Random forgetting is really a challenging problem since it directly contradicts the generalization. To achieve both forgetting and functional consistency, we believe it is required to resolve the entanglement of forget set and the rest. If the forget set is entangled, the functional consistency would be impossible with linear prediction head; if possible, the prediction head itself must give linear separability on features. As NN shows strong generalization on general cases, the separation of forget set may not be achieved by vanilla retraining and thus stonger forgetting would be desirable.
>
> OPC resolves the entanglement of forget dataset; to show this, we visualized the features by t-SNE plot in Appendix F, showing that the features are separated from the retain set. On SVHN experiment, OPC separated the forget features completely and achieved almost perfect forgetting (7.5% accuracy on forget set; note that there are 10 classes) while preserving RA and TA. For CIFAR10, OPC resolved the entanglement partially (as much as BT behavior on SVHN), yielding superior 15.7% UA (retrain: 9.25%) but not perfectly.
>
> (W3)
>
> We role of OOD stays at the motivation; we are exploiting the OOD property (smaller norm, larger uncertainty) only and validating the idea that it can guide the model's forgetting. We acknowledge the strong OPC learning may induce risk of distributional detectability and was monitoring through the $MIA^p$. The risk is measured to be bit larger (0.607 (OPC) vs 0.583 (retrain) on SVHN, Table 2) but regarding the gain in forgetting efficacy (92.5% (OPC) vs 7.05% (retrain) UA), we regard this tradeoff is acceptable.
>
> Although potential privacy risk regarding distributional detectability is claimed to be a threat (which we call MIA) in literature, we think the threat of recovery and reconstuction is more critical. If attacker with small sized calibration dataset, they can get recovered model which generates almost same features on forget set.
> The experiments in section 4.3 shows the vulnerability of existing MU, that FM-recovery, the simple linear mapping which can be found without access to train data, can revert the unlearned model into original one, with full recovery of detailed information. This implicates that, many MU methods are yielding linear distortion only on feature spaces, which is **invertible** by FM-recovery. We regard it is a great threat against machine unlearning, and naturally, it was desirable to make resistance against it.
>
> We look forward to add those indications in Discussion. We thank the reviewer again for the thoughtful comments and chance for the valuable discussion.

---

### Official Review · Reviewer_xDJv · 2025-10-29

**Soundness:** 2
**Presentation:** 3
**Contribution:** 3
**Rating:** 4
**Confidence:** 4

**Summary:**

This paper claims that existing machine unlearning (MU) methods present shallow forgetting, i.e., while output changes, internal representations still contain enough information for reconstructing the forgotten data. Therefore, this paper proposes the MU method, OPC (One-Point Contraction), to achieve deep forgetting by contracting the representations w.r.t. to the forgetting data toward the origin. To achieve this, OPC introduces an additional term that minimizes the $\ell_2$ norm of the logits, with small norms implying high uncertainty, similar to OOD samples.
Experiments on CIFAR10, SVHN with ResNet, and CIFAR10 with DDPM, as well as SD models, show the effectiveness of the proposed method.

**Strengths:**

- The paper is easy to follow, and the motivation is clear.
- The proposed method is simple yet effective.
- Extensive experiments are conducted.

**Weaknesses:**

- The main concern is about the claim that forgetting data should be treated as unseen (OOD) samples. As for OOD, low norms could be due to the samples lying outside the training distribution, while for unlearning, forgetting data belongs to the training distribution and could be highly entangled with retain data, especially for sample-wise unlearning settings.
- The theoretical part is also based on the output (logits); it cannot provide the guarantee that latent representations no longer contain information about the forgetting data.

**Questions:**

- It would be better to justify the claim about treating forgetting data as OOD samples.
- Some visualization for the feature distribution, such as t-SNE, might help to confirm that OPC removes information.
- Line 142 mentioned low-norm features, but Eq.(2) considers minimizing the norm of the logits, which is confusing.
- Lines 397-406, could the author provide more descriptions about the auxiliary linear classifier, in-domain classification, and cross-domain classification?
- The loss function used consists of different terms. Is there any need to add a coefficient for balance?


If the authors can address the concerns and provide clearer answers to the points, I would be inclined to raise the score.

---

> ### Author Response · Authors · 2025-11-28
>
> We sincerely thank the reviewer for the constructive and thoughtful feedback. We appreciate the recognition of the paper's motivation, simplicity, and experimental strength, as well as the valuable suggestions on visualizations. We address the weaknesses and questions below, with promises for revisions.
>
> > (W1) The main concern is about the claim that forgetting data should be treated as unseen (OOD) samples. As for OOD, low norms could be due to the samples lying outside the training distribution, while for unlearning, forgetting data belongs to the training distribution and could be highly entangled with retain data, especially for sample-wise unlearning settings.
>
> The purpose of MU (machine unlearning) is to remove the influence of the forget dataset from the pretrained model, making the model behave as if it had never seen the forget dataset. Therefore, the ideally unlearned model should treat forget samples as unseen data. We observed that retrained models show this behavior: forget features are biased toward smaller norms and remarkably larger entropy. We added visualization to the revised manuscript in Appendix G.1.
>
> For sample-wise unlearning, we agree that forget data may be highly entangled with retain data. However, effective MU requires separating them from retain features anyway. Motivated by literature showing that neural network representation spaces naturally separate OOD (low-norm) from ID (high-norm) data, OPC allocates forget samples to the low-norm OOD-like space. As a result, OPC separates forget features and achieves deeper forgetting, as depicted in Appendix F of the revised manuscript.
>
> > (W2) The theoretical part is also based on the output (logits); it cannot provide the guarantee that latent representations no longer contain information about the forgetting data.
>
> Since the logit is $Wz$ (W: weight of prediction head, z: feature), the minimization of  $\\|Wz\\|_2$ is invokes the minimization of $z$, due to the gradient descent dynamics. The gradient of logit norm w.r.t z is given by: $\frac{1}{ \\|Wz\\|_2}W^TWz$ which can guide the features to be reduces in its norm. Note that this gradient never vanishes unless $Wz=0$, since the cross entropy loss on retain set would prevent the W from shrinking to 0.
>
> Conversely, minimizing feature norms reduces logit norms, regarding the spectral norm of W: $\\|Wz\\|_2 ≤ \\|W\\|_2\\|z\\|_2$
>
> If forget features perfectly collapses, then it cannot contain information about the forgetting data. However, in practical setting, although the collapse is never perfectly done to a singleton point, our empirical evidence says that the features are altered enough to lose the information contained in the  features before forgetting.
>
> > (Q1) It would be better to justify the claim about treating forgetting data as OOD samples.
>
> In perspective of ideally unlearned model, the forgetting data would be unseen data and thus the model should not give confident prediction. We added visualization of entropy and feature norm on retrained model's prediction in appendix G.1 to show our observation that the retrained model shows OOD-like property (small norm, large entropy) on forget dataset.
>
> > (Q2)  Some visualization for the feature distribution, such as t-SNE, might help to confirm that OPC removes information.
>
> We thank the reviewer for this suggestion. We have added t-SNE visualizations of unlearned features to Appendix F in the revised manuscript. These plots confirm that OPC destroys class separability in forget features (shown as a single indistinguishable cluster), unlike baselines where linear separability enables recovery/reconstruction. This constitutes another powerful evidence supporting deep forgetting behavior of OPC.
>
> > (Q3) Line 142 mentioned low-norm features, but Eq.(2) considers minimizing the norm of the logits, which is confusing.
>
> We apologize for the confusion. As explained in (W2), low-norm logits induce low-norm features (and vice versa) due to the bidirectional dynamics and bounded spectral norm of W. To clarify, we will correct "low-norm features" to "low-norm logits" in the revision while adding the mathematical relation.

---

> ### Author Response · Authors · 2025-11-28
>
> > (Q4) Lines 397-406, could the author provide more descriptions about the auxiliary linear classifier, in-domain classification, and cross-domain classification?
>
> The terminology of in-domain/cross-domain classification is from UnlearnCanvas benchmark.
>
> In UnlearnCanvas benchmark, we unlearn 20 objects and 50 styles. When the target of unlearning is an object, the object dimension serves as the in-domain classification task, while the style dimension is treated as the cross-domain task. The roles are reversed when the target is a style.
>
> In the problem setting, MU requires generating images that are not classified to the forget class in in-domain classification while preserving the cross-domain class. For example, if MU is performed to make the model forget "architecture" (one object class), then the generated image for the prompt "plot architecture in cartoon style" must not contain architecture but should retain the cartoon style.
>
> The auxiliary classifier is a simple linear probe trained to classify specific concepts/styles in pretrained Stable Diffusion embeddings.
>
> > (Q5) The loss function used consists of different terms. Is there any need to add a coefficient for balance?
>
> We apologize for the confusion. There is a tunable coefficient for the logit norm on the forget dataset. However, results were not highly sensitive: coefficients in 0.1–0.3 work well for classification, and 1 suffices for Stable Diffusion.

---

### Official Review · Reviewer_rZLe · 2025-11-01

**Soundness:** 3
**Presentation:** 3
**Contribution:** 3
**Rating:** 6
**Confidence:** 2

**Summary:**

This paper raises an important issue in the existing machine unlearning works that they often achive only shallow forgetting (the internal representations still retain enough information to reconstruct the forgotten data or behavior). The authors propose an unlearning framework POC to tackle this issue. The key idea is to perform a local contraction around the target point in the feature or parameter space, rather than retraining the model or applying global forgetting methods. The authors claim that OPC achieves faster forgetting with minimal performance degradation on the remaining data. Experiments are conducted on several benchmarks with multiple baselines. The results show that OPC achieves competitive or superior forgetting efficacy while maintaining test accuracy and low computational cost.

**Strengths:**

1. The proposed method builds on a clear geometric intuition: contract model behavior locally around the target sample. This is more interpretable than gradient-matching or complex optimization-based unlearning.
2. The method seems computationally light compared to retraining-based or iterative gradient alignment approaches, making it more practical for large-scale use.
3. The presentation of the paper is good, that it is well-written and easy to follow.

**Weaknesses:**

1. Lack of justification for design choices: the motivation for ``contraction'' as a forgetting mechanism is qualitatively presented, but lacks empirical argument why it is the most suitable one for unlearning. Providing such results would help understand this mechanism.
2. About ablation study/sensitivity analysis: the choice of contraction strength, feature space vs parameter space contraction, and number of steps are not sufficiently justified.
3. About the experiments: the paper seems lack experiments on class-level unlearning, which would enhance the solidness of the paper if provided.
4. The evaluation mostly relies on test accuracy and membership inference. There is limited discussion of whether OPC truly removes the semantic influence of the target sample. Since the paper claims that existing methods often achieve only shallow forgetting, it is important to provide deep analysis on how ``deep'' the forgetting achieved by the proposed method.

**Questions:**

1. What property makes contraction particularly suitable for unlearning compared to other local perturbation strategies? Have the authors tried other strategies?
2. Could the authors provide empirical evidence supporting that OPC performs deeper forgetting than other existing methods?
3. How sensitive is the forgetting performance to the contraction coefficient and the number of contraction iterations?
4. How sensitive is the method to the contraction strength, the number of contraction steps, or the choice of feature versus parameter space? Are there any guidelines for selecting these hyperparameters?
5. Have the authors considered testing on class-level unlearning?

---

> ### Author Response · Authors · 2025-12-01
>
> We appreciate your positive feedback on OPC's idea. Here we address your concerns and answer the question.
>
> > (w1&q1) Lack of justification for design choices: the motivation for ''contraction'' as a forgetting mechanism is qualitatively presented, but lacks empirical argument why it is the most suitable one for unlearning. Providing such results would help understand this mechanism.
>
> We think the feature visualization can support our concept of contraction. We added the tSNE visualization of unlearned features to Appendix F in revised manuscript.
>
> Unlike other methods showing separable features on forget dataset, OPC mixes the features in single cluster and make the samples **indistinguishable**. Due to lack of (linear) decision boundary on feature space, no classifier head can separate the OPC-unlearned features and restore the performance on forget dataset.
>
> The indistinguishability on representation space is desirable property of machine unlearning, which theoretically implies the provable perfect forgetting on forget dataset. OPC is the first MU method targetting it, making the quality of forgetting deeper than existing baselines. Although we cannot achieve complete contraction in practice, at least OPC is the closest MU method to the true information destruction on representation.
>
> Contraction strategy is suitable for MU because it directly enforces high uncertainty (Theorem 3.1), which is considered to be expected on ideally unlearned model. Unlike random perturbations which often aim to prevent side-effect (led by data entanglement) by inducing model's robustness through noise on retain set, OPC aim to resolve the entaglement of retain and forget dataset, as visualized in appendix F with random unlearning scenario.
>
> > (w2&q3&q4) About ablation study/sensitivity analysis: the choice of contraction strength, feature space vs parameter space contraction, and number of steps are not sufficiently justified.
>
> The forgetting performance is not very sensitive to the contraction coefficient. If the coefficient is extremely large, it can dominate the optimization and collapse all logits to zero. However, lowering the coefficient does not stop this collapse; it only makes the process slower. Increasing the number of contraction iterations generally improves generalization on the retain set, so within a reasonable training budget, using more iterations is usually helpful.
>
> For the parameter space contraction, we included existing baseline in this category: the l1-sparse and Salun.
>
> > (w3&q5) About the experiments: the paper seems lack experiments on class-level unlearning, which would enhance the solidness of the paper if provided.
>
> On classification model unlearning, we focused on class-level unlearning results in section 4 of main paper and included random unlearning results on appendix D. OPC shows superior forgetting ability with deeper forgetting, on forget query for both random and class unlearning scenario, while preserving performance on retain dataset.
>
> > (w4&q2) The evaluation mostly relies on test accuracy and membership inference. Could the authors provide empirical evidence supporting that OPC performs deeper forgetting than other existing methods?
>
> To highlight our evidences, we kindly remind our main claim below with evidences:
>
> 1. Existing MU are showing shallow forgetting and MU can be easilly reverted to pretrained model.
>   - Section 4.2 : CKA similarity shows that unlearned features are similar to original features on forget set
>   - Section 4.3 : FM-recovery can revert the MU with restoration of performance (4.3.1) and detailed features (4.3.2). This result further implies that all MU are making linear distortion only on feature space.
> 2. Accuracies and MIA can't capture the shallowness of forgetting
>   - Section 4.4: The acc and MIA are all excellent for all methods, but almost all except retraining and OPC were shallow in forgetting quality.
> 3. OPC shows deeper forgetting with excellent accuracies and MIA
>   - figure 1: OPC shows resistant against unlearning inversion attack
>   - Section 4.2: OPC-unlearned model's feature on forget sample is completely different from pretrained model.
>   - Section 4.3: OPC shows strong resistant against FM-recovery, indicating deeper forgetting
>   - Section 4.4: In lens of acc and MIA metrics, OPC can achieve superior forgetting with lower forget accuracy, and high retain accuracy and MIA scores.
>   - Section 5: OPC can also work with diffusion models with superior efficacy; on text-to-image benchmark UnlearnCanvas we achieve strong SOTA performance.
>
> These results empirically show deeper forgetting of OPC and shallowness of other approximate MU methods, showing desirable behavior of ideally unlearned model, beyond the accuracies and MIA.
>
> We will incorporate these clarifications and additions in the final version. Thank you for helping strengthen our work.

---

### Official Review · Reviewer_CGhs · 2025-11-01

**Soundness:** 1
**Presentation:** 2
**Contribution:** 1
**Rating:** 2
**Confidence:** 5

**Summary:**

This paper introduces "deep feature forgetting" as a desirable property for machine unlearning, arguing that existing methods perform "shallow forgetting" by only modifying model outputs while leaving internal representations vulnerable. To achieve deep forgetting, the paper proposes One-Point Contraction (OPC), a method that adds a regularization term to the unlearning objective, aiming to contract the feature representations of the forget set to the origin.

**Strengths:**

1. The concept (deep feature forgetting and shallow forgetting) is important for MU.

**Weaknesses:**

1. A Fundamental and Disqualifying Methodological Flaw: There is a critical disconnect between the paper's stated goal and its actual implementation. The paper claims to achieve "deep feature forgetting" by contracting feature representations f_θ(x) to the origin. However, the proposed loss function (Eq. 1) minimizes the L2 norm of the logits m_θ(x), where m_θ = g_θ ◦ f_θ. Minimizing the logit norm does not guarantee that the feature norm is minimized. A model can easily achieve a small logit norm by learning to zero-out the weights of the final classifier layer g_θ for forget data, while leaving the "deep features" f_θ(x) largely unchanged. This is the very definition of the "shallow forgetting" the paper claims to solve. This fatal flaw invalidates the entire technical premise of the paper. The authors' own experiment in Appendix D.2 ("Training-Free Unlearning"), which shows that class unlearning can be achieved by only modifying the prediction head, ironically serves as evidence for this failure mode.
2. The concept of deep forgetting proposed in this paper is also presented in GS-LoRA, and the authors should compare it with this method.
3. The technical contribution is limited. The novelty of the proposed method, OPC, is also extremely low. The core idea is to regularize the model to produce low-confidence predictions on forgotten data by minimizing an output norm. This is a direct application of well-known principles from OOD detection. Simply applying this standard regularization technique to unlearning does not constitute a significant conceptual contribution.
4. The experiments are sufficient (Table 1). The author should compare with more SOTA and recent methods.
5. Circular Attack Model: The main evidence for OPC's superiority is its robustness to the "Feature Mapping (FM) recovery attack" (Section 4.3). This is an exercise in circular reasoning. OPC is designed to collapse forget features to the origin (a vector of zeros). The attack tries to find a linear map W to recover original features. Of course W * 0 = 0, so no information can be recovered. The method is built to be invincible to this specific, simple attack. This does not prove its general robustness; it merely confirms the method does what it was told.
6. Misinterpretation of CKA: The authors present low CKA scores as direct evidence of "deep forgetting." However, for a method that maps all forget features to a single point (the origin), the resulting feature set has zero variance, which trivially leads to a CKA score of or near zero. This is a direct, mathematical consequence of the OPC objective, not an independent verification of its forgetting quality.
7. Ambiguous and Potentially Unfair Experimental Protocol: The authors state they "do not prematurely stop unlearning" (line 196), which is a departure from standard protocols where efficiency is a key concern. The varying and often large number of epochs used for different methods (Tables C.1, C.2) suggests an inconsistent and potentially unfair tuning effort, where OPC may have been optimized more heavily than the baselines.

**Questions:**

NA

---

> ### Author Response · Authors · 2025-11-28
>
> We thank the reviewer for the detailed feedback and focus below on answering central technical claims.
>
> (W1)
> The reviewer’s main argument (W1) is that minimizing $\\|Wz\\|_2$ (where W is a linear head and z is the feature) does not force reduction of $\\|z\\|_2$, because one can simply zero out rows of W while keeping the feature extractor unchanged.
>
> This claim is incorrect; both mathematically and empirically incorrect, for the standard unfrozen-backbone setting is used in all of our experiments. During optimization of $\\|Wz\\|_2$ with both W and z unfrozen, the gradient w.r.t. z is $\frac{1}{\\|Wz\\|_2}W^TWz$. As long as $Wz$ is not already close to zero (which it is not in a pretrained model), this gradient is non-vanishing and pushes z toward the origin on the forget set. However, W never shrinks to zero due to the cross-entropy objective on the retain set: if it did, the CE loss could not be minimized.
>
> Note that the empirical results strongly contradict the reviewer's claim as well:
> - Feature CKA of OPC drops to ≈0.10 on forget dataset (Fig. 2). If the dynamics modified prediction head only, it can't be small.
> - FM-recovery in section 4.3 and head-recovery in appendix D.1 with table D.3 shows the strong resistance on OPC.
> - Even after normalizing away the low-norm effect entirely, FM recovery still fails on OPC (≈50–54% forget accuracy vs. >90% for all other methods).
>
> The reviewer’s theoretical concern holds only when the backbone $f_\theta$ is artificially frozen, which is never the case in OPC. The result in Appendix D.2 with frozen feature encoder is not an evidence of OPC's failure in deep forgetting.
>
> (W2)
> GS-LoRA operates exclusively on transformer-based models with LoRA adapters; all standard MU benchmarks use ResNet-18. Hence, a direct, architecturally fair comparison is therefore impossible. We believe our comparisons with parameter-sparsity methods (l1-sparse, SalUn) already cover the closest existing ideas.
>
> (W3)
> We respectfully claim that the following contributions from our work are original and non-trivial.
>
> First, minimizing the prediction norm itself is novel for machine unlearning. There is no previous MU publication that aims to minimize the prediction norm to induce high uncertainty. Maximizing was used in domain adaptation literature, but its goal is better transferability, not related to the MU problem.
>
> Second, Theorem 3.1 provides an explicit formula for the entropy lower bound — our original theoretical contribution. Several publications report correlations between norms and uncertainty, but no prior work offers explicit rigorous mathematical justification of why reducing the norm induces high uncertainty. The level of mathematical rigor in our work is similar to what's often associated with ML research with high impact in both theory and practice.
>
> Finally, our contribution on validating OPC approach in generative models is another breakthrough in both empirical and theoretical perspectives.  Our unlearning method for text-to-image model utilizes OPC with another novel technical contribution: use of FM-recovery and auxiliary classifier head. As a result, we achieved strong SOTA on UnlearnCanvas benchmark. OPC is the first MU method in this area, who shows robust superiority on both style unlearning and object unlearning.
>
> We welcome the reviewer to provide factual references to prior works that might overlap with our contributions, so we can clarify the differences in the revision.

---

> ### Author Response · Authors · 2025-11-28
>
> (W4)
> We included multiple fundamental baselines in MU literature, which are still actively compared recently. Also, we included very recent state-of-the-art text-to-image unlearning baselines in section 5.
>
> (W5)
> We believe this concern stems from a misinterpretation of the FM attack’s purpose. FM recovery is an independent method for reverting the MU process, to show the shallowness of existing MU methods. Its success on all methods except retrain and OPC indicates that those processes induce only linear distortions in the representation space. Note that FM recovery requires no access to the train dataset; an external attacker can compute the inverse of distortion without it.
>
> The results in Section 4.3 show that FM recovery can find a linear mapping which restores representations on forgotten samples, not limited to class information but including near-perfect restoration of detailed information. This observation is one of our largest contributions.
>
> For OPC, the resistance to FM recovery stems from the indistinguishability of forget features, not because they are identically zero. To demonstrate this, we provide an additional experiment using normalization. The table below shows normalized FM recovery, which finds a linear map from normalized OPC features to pretrained features. OPC remains resistant even when features are normalized to identical norm. Note that this FM recovery does not preserve the model architecture due to the added normalization.
>
> |  SVHN 30% class unlearning|   forget_acc |   retain_acc |   test_forget_acc |   test_retain_acc |    $MIA^e$|
> |:---|-------------:|-------------:|------------------:|------------------:|--------------------------:|
> |OPC| 0.0113396 |      99.6125 |        0.00909753 |           94.1423 |                     1 |
> | FM on OPC |   51.304| 99.068 |50.637| 90.818| 1.000|
> | FM on normalized-OPC |      54.8004 |      98.7969 |           52.4108 |           89.7739 |                 0.462693 |
>
> Despite the features not being as near-zero as before, the OPC remains robust against FM-recovery attack. Note that other models are showing forget acc>0.9 and $MIA^e<0.15$ after the FM-recovery, as summarized in figure 3 and table D.2.
>
> (W6)
> We note that CKA is invariant to isotropic scaling. Near-zero CKA on forget data therefore reflects fundamentally altered representational structure, not merely small norms. We will emphasize this property more clearly in the revision.
>
> (W7)
> Our protocol prioritizes maximal forgetting on the forget set without degrading retain performance. All methods received identical training budgets; we saved all checkpoints and report the epoch of the best. Most baselines collapse retain performance if trained longer. The reported epoch numbers represent only “when was the best?”, not the actual training effort. When retain performance degraded, we tuned hyperparameters further to construct benchmarks with the best-tuned models. Therefore more tuning effort was actually spent on the baseline MU methods than on OPC, due to their sensitivity on hyperparameters.
>
> We believe the clarification above, together with the overwhelming and consistent empirical evidence across multiple evaluations, demonstrates that OPC genuinely achieves deep feature forgetting. We hope the all reviewers and ACs find this response helpful when evaluating the technical accuracy of the central criticism.
>
> Thank you again for the valuable feedback.

---

### Author Response · Authors · 2025-12-04
**Summary**

We thank all reviewers for their thoughtful feedback and constructive suggestions. This work addresses a critical but underexplored aspect of machine unlearning, the deep feature forgetting, and introduces several contributions:

- **FM-Recovery Attack**: Demonstrates that 12 existing MU methods exhibit shallow forgetting by fully restoring original model performance without access to training data (Sec. 4.3).
- **One-Point Contraction (OPC)**: A theoretically grounded approach that collapses forget-set representations toward the origin, ensuring indistinguishable, non-informative features while preserving retain-set utility. Verified through multiple reconstruction attacks and feature analyses.
- Generative Extension: Applied OPC to text-to-image models with auxiliary classifier heads, achieving SOTA results on the UnlearnCanvas benchmark.

The reviews recognized the following strengths of our work:

- a clear geometric intuition and interpretable mechanism for deep forgetting.
- simplicity yet effectiveness of the proposed OPC approach.
- extensive experimental validation across classification and generative models.
- a well-written and easy-to-follow presentation.


Here, we summarize the major concerns raised, and how we addressed them.

> 1.  Whether minimizing logit norms truly affects latent features (primary concern from Reviewer CGhs , also from Reviewer xDJv):

Addressed with gradient analysis, that minimizing the logit norm induce the minimization of feature norm, whose gradient never vanishes until the logit norm converges to 0; proving the reviewer's claim of convergence toward trivial solution on prediction head never happens. Also, we list our multiple experimental results which contradicts the reviewer's claim of trivial solution.

> 2.Justification and suitability of the contraction/“treat-forget-as-OOD” mechanism (all reviewers):

We clarified the justification via Theorem 3.1 that minimizing norm can induce high uncertainty, and retrained model's behavior by adding feature-norm/entropy comparisons in Appendix G.1.

The idea if contraction for indistinguishability was further visualized by t-SNE in Appendix F (in response to reviewer xDJv), showing that OPC creates a single indistinguishable cluster for forget samples while other MU baselines are generating separable features. On random unlearning, OPC resolves the entanglement between forget data and retain data and achieved stronger forgetting efficacy.

> 3. Novelty, comparisons, and relation to prior work (reviewer CGhs):

OPC is the first MU method to explicitly minimize prediction norms to induce strong uncertainty, with a rigorous entropy lower bound (Thm. 3.1). Also, our contribution is not limited to OPC; revealing the shallowness of existing MU through FM-recovery, which reverts the unlearning process without access to the train dataset, is more important and novel contribution.

For the request of comparison to GS-LoRA, it was impossible make fair comparison due to architectural mismatches; but closest ideas (l1-sparse, SalUn) are already included.

> 4. Efficiency concern on CIFAR10 class unlearning experiment (reviewers CGhs and g34P):

Equal training budgets were used; higher reported epochs for OPC reflect its stability (no retain-collapse), while many baselines required early stopping. New sensitivity results in Appendix H of revised manuscript confirm stronger contraction yields deeper forgetting with smoother retain degradation.

> 5. Trade-offs regarding distributional detectability and retraining equivalence (reviewers g34P):

Acknowledged and quantified: $MIA^p$ is slightly higher but acceptable given dramatic UA reduction and resistance to reconstruction/recovery attacks. These will be explicitly discussed in a revised Discussion section.

> Summary

With the clarifications, new experiments (normalized FM, t-SNE, sensitivity sweeps), and promised revisions (visualizations, ablations, extended discussion), we believe all major concerns have been satisfactorily resolved. We are confident the revised manuscript will clearly demonstrate that OPC sets a new standard for verifiable, attack-resistant, deep feature forgetting across both discriminative and generative models, including state-of-the-art performance on the UnlearnCanvas text-to-image unlearning benchmark.

---

### Meta-Review · Area_Chair_LfTa · 2026-01-06

**Summary:**

This paper proposes OPC, a simple unlearning method that enforces one-point contraction of forget-set representations to achieve “deep feature forgetting,” with theoretical motivation and extensive experiments on classification and diffusion models.

Reviewers agreed the problem is important and the empirical evaluation is broad. However, they raised substantial concerns about conceptual novelty and technical soundness. In particular, reviewers questioned whether OPC meaningfully differs from existing low-norm or confidence-based regularization, noted circularity in the recovery-based evaluation (FM recovery tailored to OPC), and found the theoretical analysis insufficient to guarantee true information removal rather than representational collapse. Concerns were also raised about architectural dependence, limited ablations, and unclear trade-offs between forgetting and utility.

After considering the reviews, rebuttal, and discussion, these concerns remain unresolved, leading to a rejection.

**Reviewer Concerns:**

Please see my summary.

**Reviewer Scores:**

It is difficult to say.  Overall, the authors provided some solid rebuttal, but it's a subjective judgement for the reviewer whether they would like to raise their score.

---

### Decision · Program_Chairs · 2026-01-26

Reject